# DATA2DECISION: A PRESCRIPTIVE ANALYTICS DATA AGENT BRIDGING ENTERPRISE INFORMATION AND OPTIMAL DECISIONS

## ABSTRACT

Enterprise business analytics has evolved from simple reporting to sophisticated decision-support systems. Prescriptive analytics, as the most advanced form of business analytics, aims to recommend optimal actions based on data, yet the field lacks standardized benchmarks that reflect real-world complexity where optimization parameters must be extracted from enterprise databases. We present `Data2Decision`, the first data agent specifically designed for database-grounded prescriptive analytics that produces mathematically optimal decisions, accompanied by `Schema2Opt`, a comprehensive benchmark simulating enterprise decision environments. `Schema2Opt` transforms SQL database schemas into optimization problems, providing complete business contexts, operational databases, and verified solutions. Our `Data2Decision` agent tackles these scenarios through a two-stage pipeline with test-time scaling: first extracting optimization parameters from databases via SQL generation, then formulating and solving optimization models with multi-modeling-language consensus, achieving end-to-end automation without manual preprocessing.

## 1 INTRODUCTION

Enterprise decision-making has undergone a fundamental transformation in the digital age (Islam, 2024; Kraus et al., 2021). As organizations generate unprecedented volumes of data, business leaders increasingly rely on data-driven approaches rather than intuition-based decision-making (McAfee et al., 2012; Davenport & Harris, 2017). This shift has driven the evolution of enterprise business analytics from simple reporting systems to sophisticated decision-support frameworks that can directly impact organizational performance and competitive advantage. However, most existing business analytics methods focus primarily on *descriptive analytics* (Hong et al., 2024; Katsogiannis-Meimarakis & Koutrika, 2023; Islam, 2024), understanding past data characteristics. Text-to-SQL systems, while powerful for data retrieval and exploration (Zhong et al., 2017; Yu et al., 2018), are fundamentally descriptive analytics tools designed to answer historical questions rather than guide future actions. In contrast, *prescriptive analytics* represents the most advanced form of business analytics, recommending optimal actions through mathematical optimization and decision modeling (Lepenioti et al., 2020; Wissuchek & Zschech, 2024). This inherent limitation of Text-to-SQL and other descriptive approaches creates a significant gap between data analysis and actionable decision-making in enterprise environments, particularly when decisions require mathematical optimization to identify provably optimal solutions.

The emerging field of Text-to-OPT has made mathematical modeling more accessible by enabling natural language interfaces to optimization solvers (Tang et al., 2023; Wang et al., 2024; Zheng et al., 2024). Despite recent advances, current approaches suffer from a critical limitation: they assume optimization parameters are explicitly provided within problem descriptions as LaTeX tables or textbook-style scenarios (AhmadiTeshnizi et al., 2024; Zhang & Luo, 2025). This oversimplification misaligns with real-world enterprise workflows (Figure 1), where OR experts must collaborate with business managers and data engineers to extract decision-relevant information from complex operational databases. While some recent work has explored prescriptive analytics tasks and data extraction from database tables, such as `InsightBench` (Sahu et al., 2025), these approaches typ-

ically provide qualitative, single-dimensional action recommendations (e.g., open an incident ticket) rather than quantitative, multi-variable optimization solutions with verifiable objective values.

Beyond methodological oversimplification, existing Text-to-OPT benchmarking datasets suffer from a data generation paradigm issue. Current datasets follow a *top-down* approach: they start with textbook optimization problems (e.g., TSP, knapsack, facility location) and create variations by adjusting parameters (Xiao et al., 2023; Yang et al., 2025; Lu et al., 2025). This paradigm confines the field to recycling known formulations rather than discovering what optimization opportunities actually exist in enterprise data. Critically, this limitation constrains the potential of even the most advanced Text-to-OPT methods. Fine-tuning approaches (Tang et al., 2023; Jiang et al., 2024), as well as recent reinforcement learning methods with verifiable rewards such as SIRL (Chen et al., 2025), are

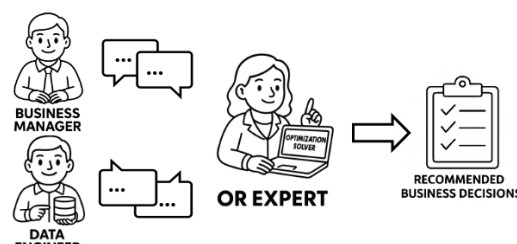

Figure 1: Real-world enterprise decision-making involves collaborative workflows where operation research (OR) experts work with business managers and data engineers to produce optimal business decisions with solvers.

bounded by this paradigm. These methods can only learn to replicate existing human optimization knowledge rather than discover novel optimization opportunities from real-world data. This raises a crucial question: *can we invert this process by starting from existing database schemas to discover latent optimization opportunities that may not exist in operations research literature?*

To bridge this gap, we present Schema2Opt, the first benchmark dataset designed with a *bottom-up* philosophy that transforms SQL database schemas into realistic optimization problems. Unlike existing benchmarks, our dataset generation pipeline discovers optimization opportunities directly from database structures. Based on this benchmark, we propose Data2Decision, the first data agent specifically designed for database-grounded prescriptive analytics that produces mathematically optimal decisions. Specifically, we make the following contributions:

- **Prescriptive Analytics Problem Definition**: We propose a realistic prescriptive analytics problem formulation that goes beyond simple descriptive analysis based on Text-to-SQL systems and addresses the practical challenges of real-world enterprise decision-making. Unlike existing Text-to-OPT approaches that assume pre-embedded parameters, our formulation requires extracting optimization parameters from enterprise databases, reflecting actual business analytics workflows.
- **Novel Bottom-up Dataset Generation Methodology**: We introduce a paradigm shift in optimization benchmarking dataset creation through Schema2Opt, which discovers optimization opportunities from database schemas rather than adapting known problems. This bottom-up approach analyzes existing information to identify latent decision variables, objective functions, and constraints, enabling the discovery of new optimization applications that may not exist in traditional operations research literature.
- **Comprehensive Benchmark Dataset**: We present Schema2Opt, a synthetic dataset specifically designed for database-grounded prescriptive analytics, along with a systematic generation framework that transforms SQL schemas into realistic optimization scenarios. Unlike existing business analytics benchmarks that provide high-level recommendations, Schema2Opt focuses on mathematical optimization problems with complete business contexts, operational databases, and verified solutions.
- **Effective Prescriptive Analytics Data Agent**: We develop Data2Decision, an effective prescriptive analytics agent that uses a two-stage pipeline with test-time scaling and multi-modeling-language consensus. This approach enables end-to-end automation without manual preprocessing and achieves up to 76.2% accuracy on Schema2Opt, demonstrating strong performance on database-grounded prescriptive analytics tasks.

## 2 RELATED WORK

### 2.1 FROM TEXT-TO-SQL TO DATA AGENTS: EVOLUTION AND LIMITATIONS

Text-to-SQL systems have evolved significantly from rule-based approaches to neural architectures (Zhong et al., 2017; Yu et al., 2018), achieving impressive accuracy on benchmarks like Spider 2.0 (Lei et al., 2024), BIRD (Li et al., 2024a), and BIRD-CRITIC (Li et al., 2025). Data agents like InsightPilot (Ma et al., 2023) and DAgent (Xu et al., 2025) extend these capabilities by automating entire analytical workflows through multi-step reasoning and cross-table associations. However, both paradigms remain fundamentally limited to descriptive and diagnostic analytics, answering what happened rather than what actions to take. While InsightBench (Sahu et al., 2025) evaluates prescriptive tasks, its recommendations remain qualitative suggestions rather than quantitative solutions with verifiable optimal values. Neither approach can formulate or solve the constrained optimization problems essential for mathematically optimal decision-making in enterprise operations where complex trade-offs require mathematical rigor. Appendix A.1 provides extended discussion of these architectural limitations.

### 2.2 TEXT-TO-OPT METHODS AND PRESCRIPTIVE ANALYTICS

The Text-to-OPT field has seen rapid development with ORLM (Tang et al., 2023) pioneering LLM use for operations research, followed by OptiMUS (Zheng et al., 2024) with iterative refinement, Chain-of-Experts (Xiao et al., 2023) with modular architectures, and SIRL (Chen et al., 2025) employing reinforcement learning with solver-based verification. However, all existing methods assume optimization parameters are embedded within problem descriptions as LaTeX tables or textbook scenarios, fundamentally misaligning with enterprise workflows where parameters must be extracted from operational databases. Furthermore, benchmarks like NL4OPT (Ramamonjison et al., 2022) and ComplexOR (Xiao et al., 2023) perpetuate a top-down paradigm, recycling textbook problems rather than discovering optimization opportunities that naturally emerge from real data structures. True prescriptive analytics requires bridging both data extraction and optimization solving (Lepenioti et al., 2020). Our work addresses this gap through `Schema2Opt`'s bottom-up approach for database-grounded prescriptive analytics, leveraging test-time scaling insights (Wang et al., 2022; Snell et al., 2024) with multi-solver consensus. Appendix A.2 examines these methods in detail, while Appendix A.3 discusses relevant test-time compute strategies.

## 3 SCHEMA2OPT: SYNTHETIC DATA GENERATION PIPELINE

Real-world prescriptive analytics fundamentally differs from traditional Text-to-OPT problems in how optimization parameters are obtained. While existing approaches assume these parameters are explicitly provided in problem descriptions, enterprise decision-making requires extracting them from operational databases through complex queries. We term this challenge **database-grounded prescriptive analytics**.

> **Definition (Database-Grounded Prescriptive Analytics).** We formalize this task as $f : (Q, S, D) \rightarrow A$, where:
> - $Q$: natural language business objectives and constraints
> - $S$: database schema with table structures
> - $D$: database content with operational data
> - $A = (x^*, v^*)$: optimal decisions and objective value
>
> The function $f$ must interpret business requirements from $Q$, identify relevant data sources through $S$, extract parameters via SQL queries against $D$, and solve the optimization model to produce $A$. Unlike traditional Text-to-OPT where parameter extraction is merely preprocessing, the database-grounded nature makes this transformation an integral component that fundamentally shapes the optimization formulation itself.

The lack of realistic benchmarks for this task stems from fundamental data sharing constraints. Organizations accumulate vast amounts of structured data in relational databases (Nambiar & Mundra, 2022), with data-driven practices growing rapidly across industries (Brynjolfsson & McElheran,

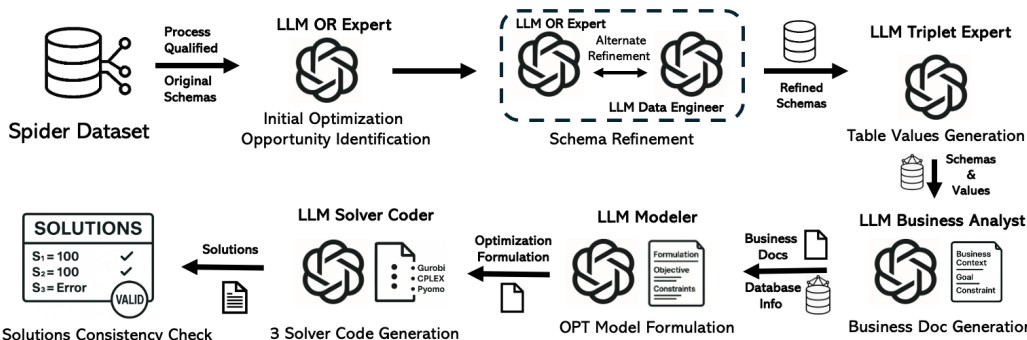

Figure 2: `Schema2Opt` synthetic data generation pipeline. Starting from Spider schemas, the framework iteratively refines schemas through OR Expert and Data Engineer dialogue, generates realistic data via Triple Expert collaboration, creates business documents, formulates optimization models, and validates solutions across multiple optimization modeling languages.

2016). However, the proprietary nature of enterprise data and embedded business logic makes sharing real-world optimization examples impossible due to governance and confidentiality requirements (Janssen et al., 2020). This creates a critical evaluation gap for systems that must bridge data extraction and optimization solving.

To address this challenge, we present `Schema2Opt`, a synthetic dataset that pairs enterprise database schemas with corresponding optimization problems. Unlike existing benchmarks that follow a top-down paradigm by starting with known optimization formulations, `Schema2Opt` introduces *the first bottom-up data generation approach that discovers optimization opportunities directly from database structures.* Our generation framework, shown in Figure 2, transforms Spider dataset schemas (Yu et al., 2018) through a five-stage pipeline. We utilize schemas from the Spider dataset with at most five tables to ensure tractability and generate two versions of the `Schema2Opt` dataset using different LLMs (`GPT-4o` and `DeepSeek-V3`) to ensure diversity and robustness. Only instances that pass multi-language majority-vote validation with consistent optimal values are included as valid cases in the final benchmark.

### 3.1 SCHEMA INITIALIZATION AND ANALYSIS

We initialize `Schema2Opt` with schemas from the Spider dataset, a widely-used SQL benchmark containing database schemas designed to resemble real-world applications, filtered to databases with at most five tables. Unlike existing benchmarks that impose predetermined optimization problems, our bottom-up approach discovers optimization opportunities by analyzing how table structures naturally map to optimization parameters.

### 3.2 ALTERNATING EXPERT DIALOGUE FOR BENCHMARK GENERATION

The core innovation of our framework lies in the alternating dialogue between specialized LLM-based agents that iteratively refine the optimization problem until convergence. Inspired by alternating optimization methods such as EM algorithms, our approach alternates between two complementary perspectives: *(1) the OR Expert agent refines optimization requirements while holding the schema fixed, then (2) the Data Engineer agent adjusts the schema to better support these requirements.* This iterative process ensures that optimization formulations and database structures co-evolve to achieve mutual consistency, similar to how alternating optimization methods achieve convergence by optimizing one component while fixing others. Appendix D provides the complete algorithmic specification and detailed prompt engineering strategies that enable this alternating optimization approach.

#### 3.2.1 OR EXPERT ANALYSIS

The OR Expert focuses exclusively on optimization modeling, evaluating how business objectives map to mathematical formulations while maintaining strict linearity constraints. Given the cur-

rent schema, the expert identifies mapping adequacy for each optimization component and specifies missing requirements without proposing database changes.

> **OR Expert Guidance:** *Focus exclusively on optimization modeling and understanding current schema-to-optimization mapping. Design linear optimization problems where objective function must be minimized/maximized $\sum$(coefficient $\times$ variable). Identify how optimization components map to current schema ...*

### 3.2.2 DATA ENGINEER IMPLEMENTATION

The Data Engineer implements schema modifications based on the OR Expert's analysis, creating tables for decision variables, adjusting columns for coefficients, and managing business configuration parameters. A key design principle is distinguishing between tabular data and scalar parameters: information that naturally forms multiple rows (e.g., product inventories, customer orders) belongs in database tables, while single-value parameters (e.g., daily capacity limits, minimum thresholds) and business formulas are stored in configuration files. This ensures proper data organization following database normalization principles.

> **Data Engineer Guidance:** *Implement schema changes following database normalization principles. Distinguish between tabular data and scalar configuration. Optimization information that represents collections belongs in database tables, while single-value parameters like capacity limits belong in configuration files. Create appropriate tables for decision variables and maintain business realism throughout the schema design ...*

This alternating process typically converges within 3-4 iterations (maximum 5), with an LLM-based judge evaluating mapping adequacy at each iteration to determine convergence, achieving complete mapping adequacy where all optimization components have identified data sources. Appendix D provides the complete algorithmic specification and detailed prompt engineering strategies that enable this alternating optimization approach.

## 3.3 DATA GENERATION AND PROBLEM FORMULATION

After schema convergence through the alternating process described in Section 3.2, Schema2Opt generates data and business descriptions. The framework employs an LLM acting as a triple expert with combined expertise in business operations, data management, and optimization modeling. This unified perspective ensures generated values reflect industry norms while maintaining cross-table consistency and enabling feasible solutions with meaningful trade-offs. An LLM-based business analyst then produces natural language descriptions that translate technical optimization requirements into business narratives, naturally incorporating configuration parameters without exposing their storage mechanism. To verify information completeness, we simulate how existing methods would solve each scenario. An LLM-based OR Expert attempts to extract optimization parameters from the business document and database to construct a complete mathematical model. Successfully producing a well-formed linear program validates that all necessary coefficients, constraints, and objectives can be derived from the provided data. Instances failing this verification are discarded, ensuring our benchmark contains only informationally complete problems. Appendix D provides detailed prompt engineering strategies for the triple expert and verification process.

## 3.4 SOLUTION GENERATION AND VERIFICATION

Schema2Opt ensures both mathematical correctness and practical solvability through a multi-stage verification process. *Template-guided generation* addresses the challenge of solver-specific API patterns that cause frequent errors in LLM-generated code. We provide templates for Gurobipy, DOCplex, and Pyomo that encapsulate best practices for variable declaration, constraint syntax, and result extraction, serving as in-context examples that significantly reduce syntax errors and API misuse. *Cross-solver validation* ensures solution reliability by executing all three solvers in parallel for each problem. We consider solutions valid only when at least two solvers agree on the optimal value within numerical tolerance or unanimously determine infeasibility. This majority voting

guards against solver-specific numerical issues while maintaining high confidence in the ground truth values for our benchmark.

## 3.5 DATASET FORMAT AND COMPONENTS

The resulting `Schema2Opt` synthetic data generation framework represents a paradigm shift in optimization benchmark creation. Rather than recycling existing textbook problems or industrial case studies, our bottom-up approach enables automatic discovery of novel optimization opportunities that emerge naturally from enterprise data patterns. Each case in `Schema2Opt` contains four core components: (1) a **business document** describing the decision scenario with context, goals and constraints, (2) a **database schema and data dictionary**, (3) **database content** and (4) **verified solutions**. We generate two dataset versions using `GPT-4o` and `DeepSeek-V3`, with 93 and 84 validated problems respectively after rigorous OR expert validation (Section G). Appendix B provides a comprehensive classification of the generated problems across business domains, optimization types, and complexity levels.

---

**Illustrative Example: Inventory Optimization**
**Business Document:** *Problem Context and Goals:* A retail company manages inventory across 3 warehouses to minimize total holding costs; *Constraints:* Daily capacity 1,000 units per warehouse, safety stock $\geq 20\%$ demand; ...
**Database Schema:** `CREATE TABLE warehouses (id INT, capacity INT, cost FLOAT); CREATE TABLE products (id INT, holding_cost FLOAT, demand INT); ...`
**Data Dictionary:** cost $\rightarrow$ storage cost per unit ...
**Database Content:** `INSERT INTO warehouses VALUES (1, 1000, 2.5), (2, 1000, 3.0); INSERT INTO products VALUES (1, 5.0, 300), (2, 7.5, 450); ...`
**Verified Solutions:** Optimal value = \$4,285.00

---

Appendix C presents six selected examples from both dataset versions that show the range of optimization scenarios discovered through our bottom-up generation method. The supplementary materials provide the complete `Schema2Opt` benchmark: all 177 validated Spider problems with business documents, database schemas, multi-solver implementations, and OR expert validation results. Each problem folder contains `problem_solution_description.md` for full documentation and `solver_execution_results.json` for verification.

## 3.6 EXTENSION TO LARGE-SCALE AND REAL-WORLD DATABASES

We extend `Schema2Opt` beyond Spider to evaluate generalization on larger, more complex databases. Table 1 summarizes the generation and validation results across three database sources: Spider provides academic schemas, BIRD (Li et al., 2024a) spans 37 professional domains with higher schema complexity, and BEAVER (Chen et al., 2024) contains production databases from MIT CSAIL infrastructure with cryptic column names and implicit relationships absent from academic benchmarks.

Table 1: Benchmark generation and validation statistics across database sources.

| Source | Generated | Solver Val. | Expert Val. | Avg Tables | Avg Rows/Table |
|--------|-----------|-------------|-------------|------------|----------------|
| Spider | 228 | 201 | 177 | 2.9 | 3.4 |
| BIRD | 138 | 120 | 2 | 4.0 | 7.5 |
| BEAVER | 12 | 11 | 6 | 4.0 | 7.2 |
| **Total** | **378** | **332** | **185** | — | — |

We achieved consistent solver validation pass rates of 87–92% across all sources, demonstrating robust scalability to larger schemas. We release 185 fully validated problems: all 177 expert-

reviewed Spider problems, 2 representative BIRD samples (`app_store`, `sales_in_weather`), and 6 BEAVER samples spanning both generators.[1]

# 4 DATA2DECISION: A PRESCRIPTIVE ANALYTICS DATA AGENT FOR ENTERPRISE DECISION-MAKING

Having established the `Schema2Opt` benchmark for database-grounded prescriptive analytics (Section 3), we now present `Data2Decision`, the first data agent specifically designed to solve these problems. Unlike existing Text-to-OPT approaches that assume pre-embedded optimization parameters, `Data2Decision` must extract these parameters from enterprise databases before solving. Our system employs a two-stage pipeline: (1) analyzing business requirements to generate SQL queries that extract decision variables, objective coefficients, and constraint parameters from databases; (2) directly transforming the SQL-enhanced problem descriptions into executable optimization code. To address inherent uncertainties in both stages, we incorporate test-time scaling through self-consistency, temperature-controlled exploration, multi-solver diversification, and majority voting consensus. This design eliminates error-prone mathematical modeling steps while ensuring robust solutions through systematic exploration of diverse formulations. Appendix E details the complete implementation, and Appendix E.3 provides the full agent instruction templates for both pipeline stages.

## 4.1 TEST-TIME SCALING THROUGH SELF-CONSISTENCY

Inspired by self-consistency approaches in reasoning tasks (Wang et al., 2022), we apply test-time scaling to prescriptive analytics by generating multiple diverse solutions and aggregating them through majority voting. This approach addresses the inherent uncertainties in database-grounded optimization: SQL queries may extract different subsets of relevant data, and the same optimization problem often admits multiple valid formulations. By sampling $N$ independent solution attempts with controlled randomness and selecting the most frequent optimal value, we significantly improve robustness over single-attempt methods.

Given $N$ parallel attempts producing optimal values, let $\mathcal{I}_{succ} \subseteq \{1, \ldots, N\}$ denote successful attempts with values $\{v_i : i \in \mathcal{I}_{succ}\}$. We employ majority voting:

$$v^* = \underset{v}{\arg\max} \, |\{i \in \mathcal{I}_{succ} : v_i = v\}|$$

where the argmax is taken over all unique values obtained. The consensus mechanism provides confidence estimates through agreement levels, with unanimous agreement indicating high confidence. We set $N = 10$ attempts per problem, which our parameter analysis indicates provides an effective balance between accuracy and efficiency.

## 4.2 TEMPERATURE-CONTROLLED EXPLORATION

We employ adaptive temperature scheduling across both pipeline stages to balance exploration and exploitation:

$$T_{sql}^{(i)} = T_{base}^{sql} + i \cdot \Delta T^{sql}, \quad T_{code}^{(i)} = T_{base}^{code} + i \cdot \Delta T^{code}$$

This progressive strategy serves different purposes at each stage. For SQL generation, temperature variation explores different interpretations of which data elements map to optimization components, addressing ambiguity when business requirements don't explicitly specify database relationships. For code generation, temperature diversity captures alternative valid formulations of the same optimization problem. In our experiments, we use wider temperature ranges for SQL generation ($T_{base}^{sql} = 0.05$, $\Delta T^{sql} = 0.05$) than code generation ($T_{base}^{code} = 0.0$, $\Delta T^{code} = 0.02$), reflecting

---

[1]The supplementary materials include the complete validated benchmark: 84 DeepSeek problems in `validated/spider_deep_84/`, 93 GPT-4o problems in `validated/spider_open_93/`, and 8 real-world samples in `validated/bird_deep/` and `validated/beaver_*/` demonstrating optimization on MIT CSAIL production systems. Each problem contains solver implementations, execution results, and expert annotations. Due to rebuttal time constraints, we will release additional BIRD and BEAVER problems after completing full expert validation. Detailed analysis appears in Appendix B.

their different uncertainty characteristics. Appendix I provides comprehensive ablation experiments analyzing different temperature scheduling strategies.

### 4.3 DIRECT OPTIMIZATION CODE GENERATION

A key design decision in `Data2Decision` is eliminating explicit mathematical modeling as an intermediate step. While conventional Text-to-OPT pipelines follow a three-stage process from natural language to mathematical formulation to executable code, we directly translate SQL-enhanced problem descriptions into solver-specific code. This approach is supported by recent findings in latent reasoning (Hao et al., 2024), which show that allowing models to reason implicitly in continuous latent space reduces hallucinations and errors compared to explicit step-by-step reasoning. We therefore bypass explicit mathematical formulation, allowing LLMs to leverage their internalized optimization knowledge directly. Our ablation study validates this design choice, showing significant performance degradation when introducing intermediate modeling steps.

### 4.4 MULTI-MODELING-LANGUAGE DIVERSIFICATION

We further enhance solution robustness by cycling through `Gurobipy`, `DOCplex`, and `Pyomo` implementations across attempts. This strategy exploits the fact that LLMs have learned distinct modeling patterns from each solver's documentation and codebase during pre-training. By leveraging these varied modeling languages, we capture diverse optimization knowledge embedded in different solver languages, enabling broader problem coverage. This multi-solver approach complements temperature-based exploration: while temperature varies problem interpretations, solver diversity accesses different pre-trained modeling patterns, preventing framework-specific limitations from affecting solution quality.

## 5 EXPERIMENTS

### 5.1 EXPERIMENTAL SETUP

**Dataset.** Two validated `Schema2Opt` test sets: `GPT-4o` (93 cases) and `DeepSeek-V3` (84 cases). All problems underwent rigorous OR expert validation, with 51 cases excluded due to quality issues and 6 ground truth values corrected (see Appendix G for details). The complete validated benchmark with exclusion criteria and correction details is available in supplementary materials under `validated/`. Pre-validation results on the full 95-case DeepSeek test set appear in Appendix F for comparison.

**Evaluation Metrics.** We focus on solution accuracy as the primary metric, measuring the percentage of problems where methods find the correct optimal solution.

**Baselines and Implementation Details.** We compare against two categories of approaches, both following a two-stage pipeline but with different setups. The `Schema2Opt` benchmark was generated using `GPT-4o` and `DeepSeek-V3`.

- **(i) Text-to-OPT methods with SQL assistance:** We evaluate `OR-LLM-Agent`, `Chain-of-Experts`, `OptiMUS 0.3`, and `ZeroShot` baseline. Since these methods cannot extract data from databases independently, we provide a **unified first stage**, using the corresponding backbone model (`GPT-4o-mini` for the `GPT-4o` test set, `DeepSeek-V3` for the `DeepSeek-V3` test set) to analyze business requirements, generate SQL queries, and extract optimization parameters from databases. This enhanced problem description containing both business context and retrieved data tables is then provided identically to all baselines for their second stage of optimization formulation.
- **(ii) End-to-end foundation models:** We evaluate `Llama-3.3-70B`, `Qwen2.5-72B`, `Phi-4`, and `Llama-4-Scout`, which handle **both stages independently**; they perform their own SQL extraction in the first stage and optimization formulation in the second stage using their respective models throughout.

**Validation Details.** We validate solutions by comparing each method's output against the ground truth optimal values from our dataset. To robustly extract results from diverse solver output formats, we use an LLM-based extraction with 5-attempt majority voting.

| Category | Method | DeepSeek-V3 Test Set (84 validated) | | GPT-4o Test Set (93 validated) | |
|---|---|---|---|---|---|
| | | Correct | Accuracy | Correct | Accuracy |
| *Text-to-OPT with SQL* | OR-LLM-Agent | 61 | 72.6% | **57** | **61.3%** |
| | ZeroShot | 54 | 64.3% | 44 | 47.3% |
| | Chain-of-Experts | 24 | 28.6% | 19 | 20.4% |
| | OptiMUS 0.3[*] | 43 | 51.2% | 3 | 3.2% |
| *End-to-End Models* | Llama-3.3-70B | 50 | 59.5% | 48 | 51.6% |
| | Qwen2.5-72B | 24 | 28.6% | 27 | 29.0% |
| | Phi-4 | 25 | 29.8% | 21 | 22.6% |
| | Llama-4-Scout | 4 | 4.8% | 1 | 1.1% |
| *Our Method* | Data2Decision | **64** | **76.2%** | 55 | 59.1% |

Table 2: Performance comparison on validated Schema2Opt benchmark after OR expert review. [*]In OptiMUS 0.3, JSON errors cannot be reliably repaired by GPT-4o-mini, causing low performance. Ground truth corrections and exclusion criteria detailed in Appendix G.

## 5.2 MAIN RESULTS

Table 2 presents performance across both validated test sets. Our Data2Decision agent achieves 76.2% accuracy on DeepSeek-V3-generated problems, the best among all methods. On GPT-4o-generated problems, Data2Decision achieves 59.1% accuracy, comparable to the best baseline OR-LLM-Agent (61.3%). Among Text-to-OPT methods with SQL assistance, OR-LLM-Agent performs strongly at 72.6% and 61.3% respectively, while Chain-of-Experts struggles at 28.6% and 20.4% despite having access to pre-extracted data. OptiMUS 0.3 shows high variance between test sets (51.2% vs 3.2%), indicating sensitivity to problem structure. End-to-end foundation models face the additional challenge of joint SQL extraction and optimization formulation. Llama-3.3-70B demonstrates competitive performance at 59.5% and 51.6%, nearly matching specialized Text-to-OPT methods despite handling the complete pipeline. Other models degrade significantly, with Llama-4-Scout achieving only 4.8% and 1.1%. Overall, Data2Decision and OR-LLM-Agent emerge as the two strongest methods, with Data2Decision excelling on DeepSeek-V3 problems and OR-LLM-Agent on GPT-4o problems.

The systematic performance gap between the two test sets across all methods reveals interesting dataset characteristics. The DeepSeek-V3-generated problems appear more amenable to optimization, with most methods achieving higher accuracy compared to their performance on GPT-4o-generated problems. This cross-dataset evaluation demonstrates the importance of diverse benchmark generation and highlights the generalization challenges in prescriptive analytics. Notably, our method maintains strong performance across both distributions, suggesting that the test-time scaling approach provides robustness beyond what single-attempt methods can achieve. Appendix H provides detailed runtime analysis and error funnel decomposition across all methods.

## 5.3 ABLATION STUDY

We conduct comprehensive ablation studies on the DeepSeek-V3 test set to analyze the contribution of each component in our Data2Decision agent (Section 4). All experiments use 10 attempts as the baseline configuration unless otherwise specified. Ablation experiments are reported on the validated subset (84 problems) after expert review, ensuring consistency with our main results in Table 2. Additional ablation experiments on temperature scheduling strategies appear in Appendix I, and runtime analysis with error funnel decomposition appears in Appendix H.

**Parameter Analysis.** Figure 3 illustrates how the number of attempts affects performance in our test-time scaling approach. The accuracy curve shows steady improvement on the validated subset (84 problems), rising from 63.1% with a single attempt to 72.6% with 3 attempts, 75.0% with 6 at-

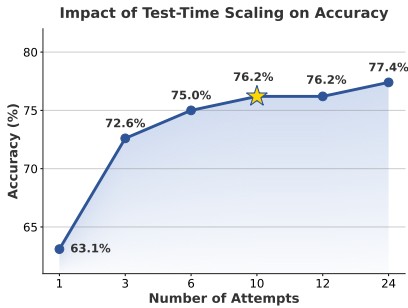

Figure 3: Impact of test-time scaling on accuracy (84 validated problems). Performance shows diminishing returns beyond 10 attempts.

tempts, and reaching 76.2% at 12 attempts. Further increasing to 24 attempts yields minimal gains (77.4%), demonstrating clear diminishing returns. Our choice of 10 attempts effectively balances solution quality with computational efficiency.

Table 3: Ablation study results on `DeepSeek-V3` validated subset (84 problems).

| Category | Configuration | Accuracy | $\Delta$ |
|---|---|---|---|
| Solver Strategy | Multi-solver consensus (full) | **76.2%** | — |
| | Single solver (Gurobi only) | 58.3% | -17.9% |
| Pipeline Design | Two-stage (full) | **76.2%** | — |
| | Three-stage | 69.0% | -7.2% |
| Temperature[†] | Incremental (ours) | **76.2%** | — |
| | Zero temperature | 73.8% | -2.4% |

[†]Detailed temperature scheduling analysis in Appendix I.

**Component Ablation.** Table 3 presents ablation results for key architectural components. (1) **Multi-solver consensus provides the largest gain (+17.9%).** Relying on a single solver (`Gurobipy`) drops accuracy from 76.2% to 58.3%, demonstrating that cross-validation across different solver implementations is crucial. (2) **Two-stage pipeline outperforms three-stage (+7.2%).** Introducing an intermediate mathematical modeling stage reduces accuracy to 69.0%, revealing that additional abstraction increases error propagation. (3) **SQL diversity matters more than code diversity.** Detailed analysis in Appendix I reveals an important insight: SQL-only incremental temperature achieves the highest accuracy (81.0%), substantially outperforming code-only incremental (79.8%). This indicates that in database-grounded prescriptive analytics, the primary uncertainty lies in data extraction interpretation rather than optimization formulation—a finding that can guide future system design. As shown in Figure 3, parallel attempts exhibit diminishing returns beyond k=6, justifying our choice of k=10 to balance accuracy and efficiency.

## 6 CONCLUSION

In this paper, we introduced database-grounded prescriptive analytics tasks, where optimization parameters must be extracted from enterprise databases rather than being explicitly provided. To address this task, we presented `Schema2Opt` (Sec 3), a comprehensive benchmark for this problem, and proposed `Data2Decision` (Sec 4), a two-stage data agent system with test-time scaling that achieves strong performance on both test sets. Our bottom-up data generation approach represents a paradigm shift from recycling textbook problems to discovering optimization opportunities inherent in database structures. This work bridges the gap between descriptive analytics and mathematical optimization, enabling end-to-end prescriptive analytics for enterprise decision-making. Our ablation analysis reveals that data extraction uncertainty dominates optimization formulation uncertainty in this setting, suggesting that future work should prioritize robust SQL generation strategies to further improve performance.

## ETHICS STATEMENT

All source data originates from publicly available academic benchmarks: Spider (Yu et al., 2018) provides database schemas with no personal information, BIRD (Li et al., 2024a) extends this with anonymized professional databases, and BEAVER (Chen et al., 2024) derives from rigorously anonymized MIT production systems. Our framework generates entirely synthetic business scenarios and numerical data without utilizing real enterprise information, avoiding privacy concerns inherent in actual business analytics (Janssen et al., 2020). We acknowledge that automated optimization could be misused for decisions affecting people without oversight; practitioners should treat our methods as decision support tools rather than autonomous decision makers.

## REPRODUCIBILITY STATEMENT

We provide all code for `Schema2Opt` generation and `Data2Decision` agent in supplementary materials. The benchmark underwent rigorous validation: 378 problems generated (228 Spider, 138

BIRD, 12 BEAVER), 332 passed multi-solver consensus, and 185 received full OR expert review (177 Spider + 2 BIRD + 6 BEAVER samples). Six ground truth values were corrected during expert validation. Each problem includes business documents, database schemas, solver implementations (`Gurobipy`, `DOCplex`, `Pyomo`), and validation results. All baseline experiments use OpenRouter API as the unified provider with Gurobi 12.0.2 and CPLEX 22.1.2 solvers. Algorithmic details appear in Appendix D and implementation details in Appendix E.

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

# Appendices

## A EXTENDED RELATED WORK

### A.1 FROM TEXT-TO-SQL TO DATA AGENTS: EVOLUTION AND LIMITATIONS

Text-to-SQL systems have evolved significantly from rule-based approaches to neural architectures (Zhong et al., 2017; Yu et al., 2018), with recent advances leveraging LLMs for improved semantic parsing (Hong et al., 2024; Katsogiannis-Meimarakis & Koutrika, 2023; Chen et al., 2020). These systems have achieved impressive accuracy on benchmarks like Spider 2.0 (Lei et al., 2024), BIRD (Li et al., 2024a), and BIRD-CRITIC (Li et al., 2025), enabling natural language interfaces for database querying. Building upon these foundations, data agents represent the next evolutionary step in automated data analysis. Systems like InsightPilot (Ma et al., 2023) pioneered LLM-empowered data exploration with goal-oriented querying capabilities. DAgent (Xu et al., 2025) advances this further by generating comprehensive analytical reports through multi-step reasoning and cross-table associations. Interactive approaches such as Tapilot-Crossing (Li et al., 2024b) incorporate self-reflection strategies to evolve agent capabilities, while InsightLens (Weng et al., 2024) integrates code, visualization, and natural language for multi-modal insight management. These agents substantially extend Text-to-SQL capabilities by automating the entire analytical workflow rather than just query generation.

However, both Text-to-SQL systems and data agents remain fundamentally limited to descriptive and diagnostic analytics. They excel at answering what happened and why it happened, but cannot determine optimal actions through mathematical optimization. While recent benchmarks like InsightBench (Sahu et al., 2025) evaluate agents on prescriptive tasks, the resulting recommendations remain qualitative suggestions such as "open an incident ticket" rather than quantitative solutions with verifiable optimal values. Neither approach can formulate or solve the constrained optimization problems essential for mathematically optimal decision-making in enterprise operations.

### A.2 TEXT-TO-OPT METHODS AND PRESCRIPTIVE ANALYTICS

The Text-to-OPT field has seen rapid development with various approaches addressing mathematical optimization from natural language. ORLM (Tang et al., 2023) pioneered the use of LLMs for optimization modeling in operations research, while subsequent methods like Chain-of-Experts (Xiao et al., 2023) introduced multi-component frameworks with domain-specific modules orchestrated by a conductor. OptiMUS (Zheng et al., 2024) further advanced the field with a modular system capable of iterative model refinement and debugging. LLMOPT (Jiang et al., 2024) proposes a unified learning-based framework with a five-element formulation. OR-LLM-Agent (Zhang & Luo, 2025) leverages reasoning LLMs through task decomposition into several stages. Autoformulation (Astorga et al., 2024) introduces Monte Carlo Tree Search to systematically explore the formulation space, while SIRL (Chen et al., 2025) employs reinforcement learning with solver-based verification. Recent work on OptiTrust (Lima et al., 2025) introduces verifiable synthetic data generation pipelines for training trustworthy optimization modeling agents.

Despite these advances, all existing Text-to-OPT methods share a critical limitation: they assume that all necessary data is embedded within problem descriptions, typically in the form of LaTeX

tables or natural language specifications. This assumption fundamentally misaligns with real-world enterprise scenarios where optimization parameters must be extracted from large databases. Furthermore, existing benchmarks like NL4OPT (Ramamonjison et al., 2022), ComplexOR (Xiao et al., 2023), and IndustryOR (Huang et al., 2025) perpetuate a problematic top-down data generation paradigm. These datasets are created by starting with well-known optimization problems from textbooks or literature, then generating variations through parameter adjustments or constraint modifications. This approach confines the field to recycling existing formulations rather than discovering optimization opportunities that naturally emerge from enterprise data structures.

True prescriptive analytics requires bridging this gap between data extraction and optimization solving (Lepenioti et al., 2020). Unlike Text-to-SQL which retrieves data from databases, or Text-to-OPT which assumes pre-extracted parameters, prescriptive analytics must accomplish both: extracting optimization parameters from databases and solving for optimal decisions. While recent solutions like PresAIse (Sun et al., 2024) combine causal inference with optimization, and Insight-Bench (Sahu et al., 2025) evaluates prescriptive tasks, these approaches provide only high-level suggestions. This gap motivates our development of `Schema2Opt` for rigorous evaluation of database-grounded prescriptive analytics.

## A.3 TEST-TIME SCALING FOR COMPLEX REASONING

Recent work has shown that Large Language Model (LLM) performance can be improved through test-time scaling, such as generating intermediate reasoning steps (Wei et al., 2022; Yao et al., 2023; Besta et al., 2024). Self-consistency (Wang et al., 2022) also shows that sampling multiple reasoning paths and selecting the most frequent answer enhances reliability on complex reasoning tasks. More recent advances demonstrate that test-time compute scaling can be more effective than simply increasing model parameters (Snell et al., 2024). Methods like Forest-of-Thought (Bi et al., 2024) extend tree-based reasoning to explore multiple solution paths simultaneously, enabling backtracking and lookahead capabilities. Process-based reward models have shown particular promise in verifying intermediate reasoning steps (Zhang et al., 2025), improving both the quality and reliability of generated solutions. These approaches have proven particularly effective for mathematical reasoning and code generation, with compute-optimal strategies achieving over 4x efficiency improvements compared to best-of-N baselines.

## B DATASET CLASSIFICATION AND ANALYSIS

This appendix provides comprehensive statistics and classification analysis for the complete `Schema2Opt` benchmark. We generated optimization problems from three database sources using two LLMs, resulting in six dataset variants. Table 4 summarizes the generation and validation results across all datasets.

| Dataset | Generated | Solver Val. | Expert Val. | Avg Tables | Avg Rows/Table |
|---|---|---|---|---|---|
| Spider Deep | 114 | 95 | 84 | 3.1 | 3.3 |
| Spider Open | 114 | 106 | 93 | 2.7 | 3.4 |
| BIRD Deep | 69 | 57 | 2 | 4.7 | 7.0 |
| BIRD Open | 69 | 63 | — | 3.3 | 8.0 |
| BEAVER Deep | 6 | 6 | 3 | 4.3 | 6.7 |
| BEAVER Open | 6 | 5 | 3 | 3.7 | 7.6 |
| **Total** | **378** | **332** | **185** | — | — |

Table 4: Benchmark statistics across all datasets. Generated: initial LLM output. Solver Val.: passed multi-solver consensus. Expert Val.: received full OR expert manual review (185 total released). We will release additional BIRD/BEAVER problems after completing full expert validation.

The table reveals several patterns. First, multi-solver consensus filters approximately 12% of generated problems across all sources. Second, schema complexity increases from Spider to BIRD and BEAVER, with average table counts rising from 2.9 to 4.0 and data density more than doubling. Third, the generation pipeline scales effectively to larger databases, with BIRD problems involving up to 10 tables compared to Spider's maximum of 6.

## B.1 SPIDER DATASET ANALYSIS

The Spider-based datasets form the core of `Schema2Opt`, providing 177 validated problems after expert review. To understand the characteristics of these problems, we employed GPT-5.1-mini to classify each instance across five dimensions: business domain, optimization type, problem complexity, implementation difficulty, and real-world applicability score.

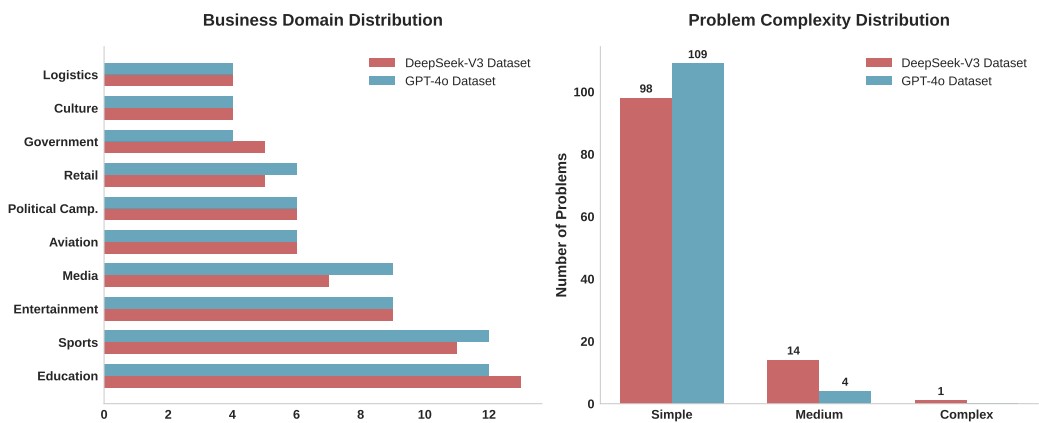

Figure 4: Spider dataset: Business domain distribution and problem complexity analysis across `DeepSeek-V3` and `GPT-4o` generated datasets.

The classification results reveal distinct patterns in problem generation. Our analysis identified 37 unique business sectors across both datasets, including education (25), sports (23), entertainment (18), media (16), aviation (12), political campaigns (12), retail (11), government (9), culture (8), logistics (8), manufacturing (8), hospitality (7), retail banking (6), college athletics (6), finance (5), technology (5), events and wedding planning (4), maritime shipping (4), academic conferences (3), academic institutions (3), disaster response (3), agriculture (2), construction (2), energy (2), gaming (2), healthcare (2), research (2), winemaking (2), and several specialized domains appearing once such as customer service centers, performing arts production, streaming systems, and transportation operations. Figure 4 visualizes the top 10 domains, showing how education, sports, and entertainment lead the distribution for both `DeepSeek-V3` and `GPT-4o` datasets.

The complexity analysis reveals a strong preference for tractable problems. With 86.7% of `DeepSeek-V3` problems and 96.5% of `GPT-4o` problems classified as simple, our bottom-up generation approach appears to naturally discover linear programming formulations that are computationally feasible. Only 14 `DeepSeek-V3` problems and 4 `GPT-4o` problems reach medium complexity, while a single complex problem appears in the `DeepSeek-V3` dataset. This distribution suggests that database schemas inherently encode optimization opportunities that align well with standard solver capabilities.

Figure 5 illustrates another consistent pattern: both datasets favor maximization over minimization objectives. The `DeepSeek-V3` dataset contains 67.3% maximization problems while `GPT-4o` reaches 75.2%, reflecting how businesses naturally frame goals around maximizing revenue, efficiency, or satisfaction. This preference aligns with common enterprise objectives such as profit maximization, throughput optimization, and customer satisfaction improvement. After rigorous validation, 84 problems from the `DeepSeek-V3` generation and 93 from `GPT-4o` passed all verification criteria and OR expert review to form the final Spider benchmark.

## B.2 BIRD DATASET ANALYSIS

The BIRD benchmark provides an opportunity to test whether our bottom-up generation approach (Section 3) scales to larger and more professionally oriented databases. This extension addresses a key question for database-grounded prescriptive analytics: can methods trained on academic

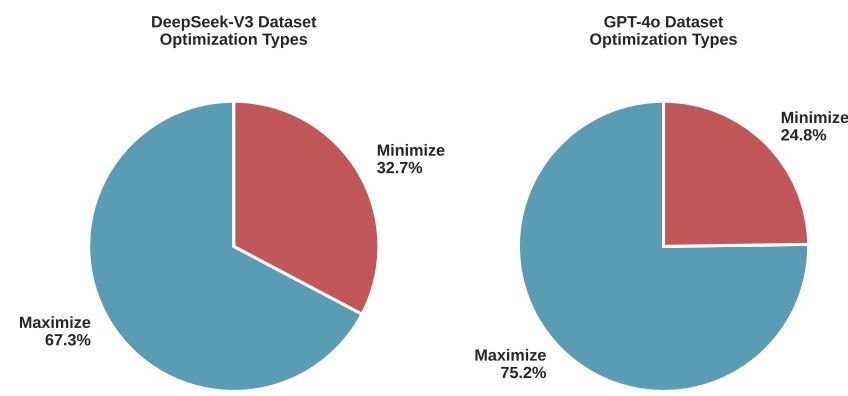

Figure 5: Spider dataset: Distribution of optimization objectives showing preference for maximization problems in both datasets.

schemas generalize to enterprise-scale databases with more complex relationships? BIRD contains 95 databases spanning 37 professional domains including financial services, healthcare administration, and municipal government. These databases are substantially larger than Spider, with more tables, more rows per table, and more complex relationships between entities.

We applied the same generation pipeline to BIRD, producing 138 candidate problems with 69 from each LLM. Using the same classification framework as Spider, we analyzed the resulting problem characteristics.

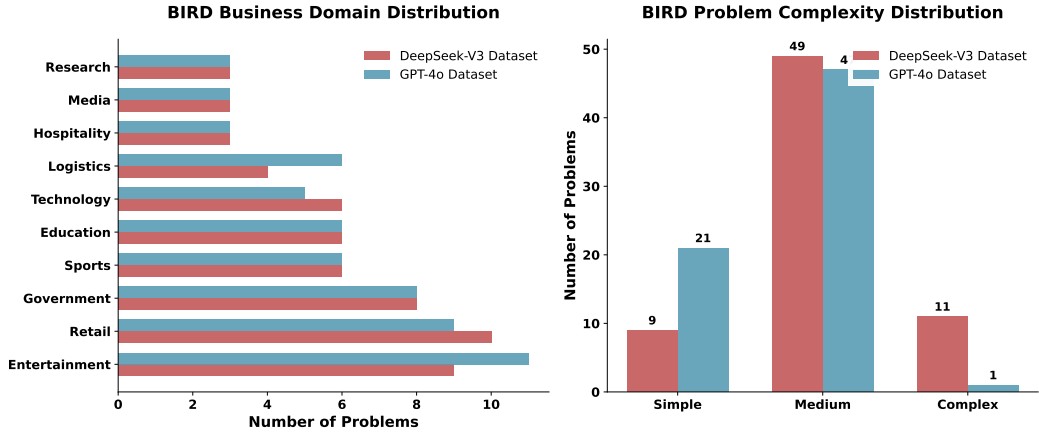

Figure 6: BIRD dataset: Business domain distribution and problem complexity analysis across `DeepSeek-V3` and `GPT-4o` generated datasets.

Figure 6 reveals that BIRD problems concentrate in different domains than Spider. Entertainment leads with 20 problems (14.5%), followed by retail with 19 problems (13.8%) and government with 16 problems (11.6%). This distribution reflects the professional nature of BIRD's source databases, which include retail analytics systems, municipal data warehouses, and entertainment industry databases.

The most striking difference from Spider appears in the complexity distribution. While Spider problems are predominantly simple (over 90%), BIRD problems shift dramatically toward medium complexity. In the `DeepSeek-V3` generation, 69.6% of problems reach medium complexity compared to only 13.3% in Spider. The `GPT-4o` generation shows a similar pattern with 68.1% medium complexity. This shift makes sense given the larger schemas: more tables create more opportunities for multi-table constraints and compound objectives that increase formulation complexity. The `DeepSeek-V3` model also generates 11 complex problems (15.9%) from BIRD schemas, com-

pared to just 1 (1.4%) from `GPT-4o`, suggesting that DeepSeek-V3 more aggressively explores sophisticated optimization opportunities when given richer data structures.

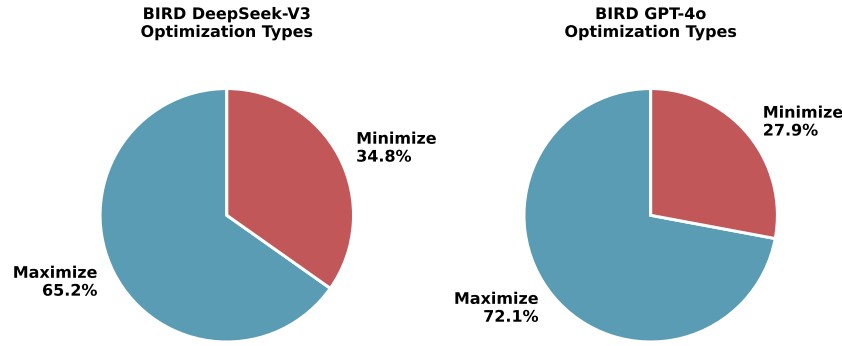

Figure 7: BIRD dataset: Distribution of optimization objectives.

Despite these complexity differences, BIRD problems maintain the same preference for maximization objectives observed in Spider (Figure 7). The `DeepSeek-V3` generation produces 65.2% maximization problems while `GPT-4o` reaches 72.0%. This consistency across database sources suggests the pattern reflects fundamental business optimization needs rather than artifacts of particular schemas.

### B.3 BEAVER DATASET ANALYSIS

BEAVER represents the most challenging test of our generation approach and the ultimate validation of database-grounded prescriptive analytics. Unlike Spider and BIRD, which contain databases designed for academic benchmarking, BEAVER derives from actual MIT CSAIL production infrastructure. These databases include operational systems for data warehousing (`dw`), block storage allocation (`csail_stata_cinder`), virtual network management (`csail_stata_neutron`), and VM image distribution (`csail_stata_glance`)—each representing real optimization scenarios with cryptic column names reflecting legacy systems, implicit relationships not captured in foreign key constraints, and domain-specific conventions requiring operational knowledge to interpret. Success on BEAVER demonstrates that `Schema2Opt`'s generation pipeline and `Data2Decision`'s solving approach can handle real-world database complexity.

We generated 12 problems from BEAVER schemas, with 6 from each LLM. The validated samples span query workload optimization, storage allocation optimization, network resource optimization, and image distribution optimization. Given the small sample size, we focus on qualitative patterns rather than precise percentages.

Figure 8 shows that manufacturing dominates the BEAVER problems, accounting for 66.7% of the generated scenarios. This concentration reflects BEAVER's origin in MIT's operational databases for production systems. All 12 problems are classified as simple or medium complexity, which is a practical outcome for production environments where optimization must complete reliably without excessive computational burden.

The most notable pattern in BEAVER is the shift toward minimization objectives (Figure 9). While Spider and BIRD both favor maximization by roughly 70%, BEAVER shows different tendencies by model. The `DeepSeek-V3` generation produces a balanced 50%/50% split between maximization and minimization, while `GPT-4o` generates 100% minimization problems. This shift likely reflects the manufacturing context where cost minimization, waste reduction, and resource conservation are primary concerns.

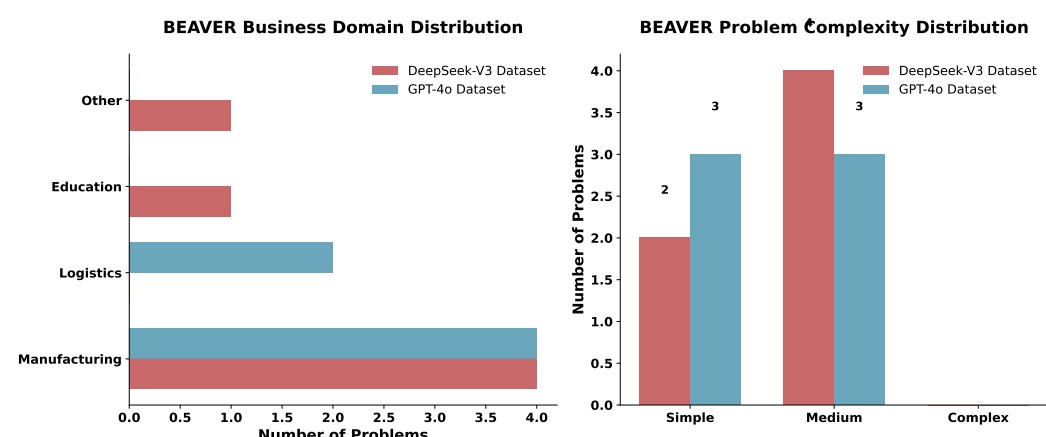

Figure 8: BEAVER dataset: Business domain distribution and problem complexity.

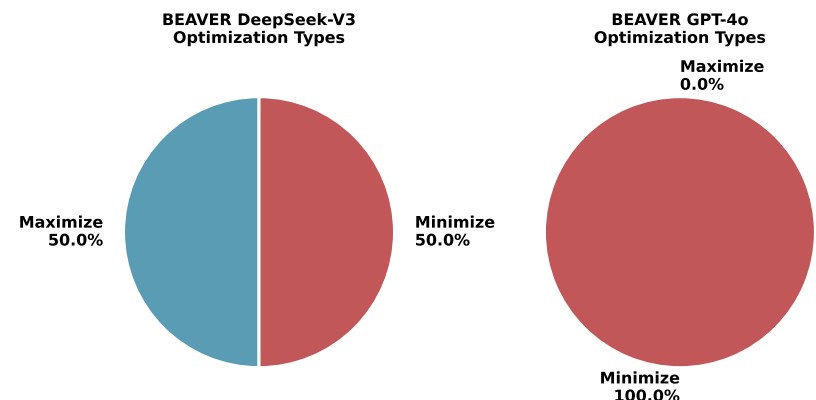

Figure 9: BEAVER dataset: Optimization type distribution.

### B.4 CROSS-DATASET COMPARISON

This section synthesizes findings across all three database sources to validate the core premise of Schema2Opt: that our bottom-up generation approach discovers meaningful optimization opportunities regardless of schema complexity or domain characteristics. Table 5 summarizes the key differences across all three database sources, revealing how database structure influences the types and complexity of optimization problems that emerge.

| Metric | Spider | BIRD | BEAVER |
|---|---|---|---|
| Generated Problems | 228 | 138 | 12 |
| Solver Validated | 201 | 120 | 11 |
| Expert Validated | **177** | 2 | 6 |
| Unique Domains | 37 | 15 | 4 |
| Avg Tables/Problem | 2.9 | 4.0 | 4.0 |
| Simple Complexity | 91.5% | 21.7% | 41.7% |
| Medium Complexity | 7.5% | 69.6% | 58.3% |
| Complex | 1.0% | 8.7% | 0% |
| Maximization | 71.1% | 68.1% | 25.0% |

Table 5: Cross-dataset comparison. Generated: initial LLM output. Solver Validated: passed multi-solver consensus. Expert Validated: received full OR expert manual review (185 total released for benchmarking).

Several patterns emerge from this comparison. First, schema complexity correlates with problem complexity. Spider's smaller schemas produce mostly simple optimization problems, while BIRD's larger schemas shift the distribution toward medium and complex formulations. Second, domain characteristics influence objective types. Manufacturing-oriented BEAVER databases favor minimization objectives for cost and waste reduction, while the diverse Spider and BIRD domains favor profit and revenue maximization. Third, the generation pipeline maintains consistent validation rates across all sources, suggesting robust scalability to different database types.

These findings validate the core premise of `Schema2Opt`: our bottom-up generation approach successfully discovers optimization opportunities across schema complexity levels while maintaining solver tractability. The 332 validated problems span a range of difficulty levels suitable for benchmarking optimization agents at different capability tiers.

## C    REPRESENTATIVE DATASET EXAMPLES FROM SCHEMA2OPT

We present six complete examples from `Schema2Opt` that illustrate the breadth of optimization scenarios our bottom-up generation approach discovers. These examples, selected from both the `GPT-4o` and `DeepSeek-V3` generated datasets, showcase how database schemas naturally encode diverse business decision problems spanning education, sports, entertainment, and logistics domains. Each example demonstrates the complete transformation from an existing SQL schema through our alternating expert dialogue to a verified optimization solution.

The examples reveal several key characteristics of our generation framework. First, they show how the same underlying database structure can support different optimization objectives through our iterative refinement process. For instance, Examples 1 and 2 both involve university activity data but diverge into distinct optimization problems focused on participation versus scoring. Second, they demonstrate the framework's ability to handle varying complexity levels, from simple linear programs with a handful of variables to more complex scheduling problems with multiple constraint types. Finally, these examples validate our design principle of separating tabular data from scalar configuration parameters, showing how business rules naturally emerge as configuration values while entity-specific data resides in database tables.

### C.1    EXAMPLE 1: UNIVERSITY ACTIVITY ALLOCATION (DEEPSEEK-V3-GENERATED)

This example demonstrates how student and faculty participation constraints naturally map to binary allocation decisions with capacity limitations.

---

**Problem Context, Goals and Constraints**

**Context**

A university is managing the allocation of students and faculty to extracurricular activities with the goal of maximizing overall participation. The decision-making process involves determining which students and faculty members should participate in which activities. Each student can participate in at most one activity, and each faculty member can participate in at most two activities. Additionally, each activity has a predefined maximum number of participants that cannot be exceeded.

The business configuration includes the following operational parameters: **Faculty Availability Limit**: Faculty members are limited to participating in a maximum of two activities to balance their workload and ensure availability. **Student Preference Threshold**: Students are allowed to participate in only one activity to ensure focused engagement and avoid overcommitment. **Total Participation Calculation**: The total participation in an activity is calculated as the sum of student and faculty participation in that activity.

The optimization problem is designed to ensure that these constraints are respected while maximizing the total number of participants across all activities.

**Goals**

The primary goal of this optimization problem is to maximize the total participation in extracurricular activities by both students and faculty. Success is measured by the total number of participants across all activities, which is the sum of student and faculty participation. This goal aligns with the operational parameters and ensures that the allocation respects the constraints on faculty availability, student preferences, and activity capacity limits.

**Constraints**

The optimization problem must adhere to the following constraints: 1. **Student Participation Limit**: Each student can participate in at most one activity. This ensures that students are not overcommitted and can focus on their chosen activity. 2. **Faculty Participation Limit**: Each faculty member can participate in at most two activities. This constraint balances faculty workload and ensures their availability across activities. 3. **Activity Capacity Limit**: The total number of participants in each activity, including both students and faculty, must not exceed the predefined maximum capacity for that activity. This ensures that activities are not overcrowded and can operate effectively.

These constraints are designed to ensure that the allocation of participants is feasible and aligns with the operational capabilities of the university.

---

## Available Data - Database Schema

```
CREATE TABLE Participates_in (
    stuid INTEGER,
    actid INTEGER
);

CREATE TABLE Faculty_Participates_in (
    FacID INTEGER,
    actid INTEGER
);

CREATE TABLE Activity_Capacity (
    actid INTEGER,
    max_participants INTEGER
);

INSERT INTO Participates_in (stuid, actid) VALUES (101, 1);
INSERT INTO Participates_in (stuid, actid) VALUES (102, 2);
INSERT INTO Participates_in (stuid, actid) VALUES (103, 3);

INSERT INTO Faculty_Participates_in (FacID, actid) VALUES (201, 1);
INSERT INTO Faculty_Participates_in (FacID, actid) VALUES (202, 2);
INSERT INTO Faculty_Participates_in (FacID, actid) VALUES (203, 3);

INSERT INTO Activity_Capacity (actid, max_participants) VALUES (1, 10);
INSERT INTO Activity_Capacity (actid, max_participants) VALUES (2, 15);
INSERT INTO Activity_Capacity (actid, max_participants) VALUES (3, 20);
```

**Data Dictionary** - **Participates_in**: Tracks student participation in activities. - **stuid**: Unique identifier for a student. Used to determine which students participate in which activities. - **actid**: Unique identifier for an activity. Used to link students to specific activities. - **Faculty_Participates_in**: Tracks faculty participation in activities. - **FacID**: Unique identifier for a faculty member. Used to determine which faculty members participate in which activities. - **actid**: Unique identifier for an activity. Used to link faculty members to specific activities. - **Activity_Capacity**: Defines the maximum number of participants allowed in each activity. - **actid**: Unique identifier for an activity. Used to link capacity limits to specific activities. - **max_participants**: The maximum number of participants allowed in the activity. Used to enforce capacity constraints.

## Mathematical Optimization Formulation

**Decision Variables**
- $x_{s,a}$: Binary variable indicating whether student $s$ participates in activity $a$ (1 if yes, 0 otherwise).
- $y_{f,a}$: Binary variable indicating whether faculty $f$ participates in activity $a$ (1 if yes, 0 otherwise).

**Objective Function**
Maximize the total participation across all activities:

$$\text{Maximize} \sum_s \sum_a x_{s,a} + \sum_f \sum_a y_{f,a}$$

**Coefficients**: All coefficients are 1, as each participant (student or faculty) contributes equally to the total participation.

**Constraints**

$$\text{1. Student Participation Limit:} \sum_a x_{s,a} \leq 1 \quad \forall s \tag{1}$$

$$\text{2. Faculty Participation Limit:} \sum_a y_{f,a} \leq 2 \quad \forall f \tag{2}$$

$$\text{3. Activity Capacity Limit:} \sum_s x_{s,a} + \sum_f y_{f,a} \leq \text{max\_participants}_a \quad \forall a \tag{3}$$

$$\text{4. Binary Constraints:} x_{s,a} \in \{0, 1\} \quad \forall s, a; \quad y_{f,a} \in \{0, 1\} \quad \forall f, a \tag{4}$$

## Cross-Solver Analysis and Final Recommendation

**Solver Results Comparison**

| Solver | Status | Optimal Value | Execution Time |
|---|---|---|---|
| Gurobipy | OPTIMAL | 9.00 | 0.18s |
| Docplex | OPTIMAL | 9.00 | 1.13s |
| Pyomo | OPTIMAL | 9.00 | 0.94s |

**Solver Consistency Analysis**
Result: All solvers produced consistent results ✓
Consistent Solvers: gurobipy, docplex, pyomo
Majority Vote Optimal Value: 9.0
**Final Recommendation**
Recommended Optimal Value: 9.0
Confidence Level: HIGH

Preferred Solver(s): gurobipy
Reasoning: Gurobipy is recommended due to its fastest execution time while still providing the optimal solution.

## C.2 EXAMPLE 2: FACULTY ACTIVITY OPTIMIZATION (GPT-4o-GENERATED)

Here the framework (GPT-4o as the backend LLM) transforms the same domain into a scoring optimization problem, showing how different business objectives emerge from similar database structures through our iterative refinement process.

### Problem Context, Goals and Constraints

**Context**
The university is focused on optimizing the allocation of faculty members to various activities to enhance the overall participation scores. The decision-making process involves determining whether a faculty member, identified by their unique ID, should be assigned to a specific activity. This decision is represented by binary variables, where each variable indicates if a faculty member is assigned to an activity. The primary objective is to maximize the total participation score, which is calculated by summing the product of participation scores for each faculty-activity pair and the corresponding binary decision variable.
Operational parameters are crucial in this context. Each faculty member has a maximum number of activities they can participate in, ensuring they are not overburdened. Additionally, each activity requires a minimum number of faculty members to ensure it is adequately staffed. These parameters are derived from the business configuration, which includes the maximum number of activities a faculty member can participate in and the minimum number of faculty members required for an activity. The problem is structured to ensure that these constraints are respected, leading to a linear optimization formulation.

**Goals**
The primary goal of this optimization problem is to maximize the total participation score. This involves assigning faculty members to activities in a way that the sum of the participation scores for all faculty-activity assignments is maximized. The success of this optimization is measured by the total participation score achieved, which directly correlates with the participation scores assigned to each faculty-activity pair. The goal is articulated in natural language to emphasize the linear nature of the optimization objective.

**Constraints**
The optimization problem is subject to several constraints that ensure the feasibility and practicality of the solution: - Each faculty member can participate in a limited number of activities, as defined by their availability. This constraint ensures that the sum of the binary decision variables for each faculty member does not exceed their maximum availability. - Each activity must have a minimum number of faculty members assigned to it. This constraint ensures that the sum of the binary decision variables for each activity meets or exceeds the required staffing level.

### Available Data - Database Schema

```
CREATE TABLE Participation_Score (
    FacID INTEGER,
    actid INTEGER,
    participation_score FLOAT
);

CREATE TABLE Faculty_Participates_in (
    FacID INTEGER,
    actid INTEGER,
    participation_score FLOAT
);

INSERT INTO Participation_Score (FacID, actid, participation_score) VALUES (1, 101, 12.0);
INSERT INTO Participation_Score (FacID, actid, participation_score) VALUES (2, 102, 18.5);
INSERT INTO Participation_Score (FacID, actid, participation_score) VALUES (3, 103, 14.0);

INSERT INTO Faculty_Participates_in (FacID, actid, participation_score) VALUES (1, 101, 12.0);
INSERT INTO Faculty_Participates_in (FacID, actid, participation_score) VALUES (2, 102, 18.5);
INSERT INTO Faculty_Participates_in (FacID, actid, participation_score) VALUES (3, 103, 14.0);
INSERT INTO Faculty_Participates_in (FacID, actid, participation_score) VALUES (1, 102, 10.0);
INSERT INTO Faculty_Participates_in (FacID, actid, participation_score) VALUES (2, 103, 16.0);
```

**Data Dictionary**
The data dictionary provides a comprehensive mapping of tables and columns to their business purposes and optimization roles:
- **Participation_Score Table**: This table stores the participation scores for each faculty-activity pair. The participation score represents the benefit of assigning a specific faculty member to an activity. The table includes: - **FacID**: Represents the unique identifier for each faculty member. - **actid**: Represents the unique identifier for each activity. - **participation_score**: Represents the score associated with assigning a faculty member to an activity, serving as a coefficient in the objective function.
- **Faculty_Participates_in Table**: This table tracks the participation of faculty members in activities. It includes: - **FacID**: Represents the unique identifier for each faculty member. - **actid**: Represents the unique identifier for each activity. - **participation_score**: Although included, this column is primarily used for tracking purposes and aligns with the participation scores in the Participation_Score table.

---

### Mathematical Optimization Formulation

**Decision Variables** - Let $x_{i,j}$ be a binary decision variable where $x_{i,j} = 1$ if faculty member $i$ is assigned to activity $j$, and $x_{i,j} = 0$ otherwise.

**Objective Function**
Maximize the total participation score:

$$\text{Maximize} \sum_{i,j} \text{Participation\_Score}(i,j) \times x_{i,j}$$

Where: - Participation_Score$(i,j)$ is the participation score for faculty member $i$ and activity $j$ from the `Participation_Score` table.

**Constraints**

1. Faculty Availability Constraint: Each faculty member can participate in a limited number of activities. $\quad(5)$

$$\sum_{j} x_{i,j} \leq \text{Max\_Activities}(i) \quad \forall i \tag{6}$$

- Max_Activities$(i)$ is the maximum number of activities faculty member $i$ can participate in, $\quad(7)$
derived from business configuration. $\quad(8)$

2. Activity Staffing Constraint: Each activity must have a minimum number of faculty members assigned. $\quad(9)$

$$\sum_{i} x_{i,j} \geq \text{Min\_Faculty}(j) \quad \forall j \tag{10}$$

- Min_Faculty$(j)$ is the minimum number of faculty members required for activity $j$, $\quad(11)$
derived from business configuration. $\quad(12)$

3. Binary Constraint: Each decision variable is binary. $\quad(13)$

$$x_{i,j} \in \{0,1\} \quad \forall i,j \tag{14}$$

Data Source Verification: - Participation scores Participation_Score$(i,j)$ are sourced from the `Participation_Score` table.
- Maximum activities per faculty Max_Activities$(i)$ and minimum faculty per activity Min_Faculty$(j)$ are derived from business configuration parameters, which are not explicitly detailed in the provided data but are assumed to be part of the business rules. This formulation provides a complete, immediately solvable linear mathematical model using the given data and constraints.

---

### Cross-Solver Analysis and Final Recommendation

**Solver Results Comparison**

| Solver | Status | Optimal Value | Execution Time |
|---|---|---|---|
| Gurobipy | ERROR | N/A | 0.18s |
| Docplex | OPTIMAL | 70.50 | 1.39s |
| Pyomo | OPTIMAL | 70.50 | 1.12s |

**Solver Consistency Analysis**
Result: All solvers produced consistent results ✓
Consistent Solvers: docplex, pyomo
Majority Vote Optimal Value: 70.5
**Final Recommendation**
Recommended Optimal Value: 70.5
Confidence Level: HIGH
Preferred Solver(s): docplex/pyomo
Reasoning: Both DOCplex and Pyomo provided consistent and optimal results, indicating reliability. Gurobipy's error suggests data issues that need addressing before it can be considered.

---

The next two examples demonstrate how our framework handles optimization in the sports domain, specifically bodybuilder team selection and performance optimization. These cases show how physical attributes and performance metrics stored in separate database tables can be unified into cohesive optimization models through our schema refinement process.

## C.3   EXAMPLE 3: BODYBUILDER TEAM SELECTION (DEEPSEEK-V3-GENERATED)

This example shows the integration of performance metrics with physical constraints, where team composition must balance total scoring against average height and weight requirements.

### Problem Context, Goals and Constraints

**Context**
A bodybuilding competition organizer is tasked with selecting a team of bodybuilders to compete in an upcoming event. The goal

is to assemble a team that maximizes the total performance score based on the bodybuilders' Snatch and Clean & Jerk scores. The selection process must adhere to specific operational constraints to ensure the team meets diversity and physical criteria.

The organizer must decide which bodybuilders to include in the team, represented by a binary decision for each individual. The total number of bodybuilders in the team cannot exceed a predefined limit, ensuring the team remains manageable and diverse. Additionally, the team must meet a minimum average height requirement of 170 cm and a maximum average weight requirement of 100 kg. These constraints ensure the team aligns with the competition's physical standards.

The performance scores for each bodybuilder are derived from their Snatch and Clean & Jerk results, which are stored in the database. The physical attributes of height and weight are also recorded and used to enforce the team's physical criteria. The business configuration includes a maximum team size limit of 5 bodybuilders, a minimum average height requirement, and a maximum average weight requirement, all of which are critical to the selection process.

**Goals**

The primary goal of this optimization problem is to maximize the total performance score of the selected team. This score is calculated as the sum of the Snatch and Clean & Jerk scores of the chosen bodybuilders. Success is measured by achieving the highest possible total score while adhering to the constraints on team size, average height, and average weight.

**Constraints**

The selection of bodybuilders for the team must respect the following constraints: 1. **Team Size Limit**: The total number of bodybuilders selected for the team must not exceed the predefined limit of 5. This ensures the team remains manageable and diverse. 2. **Minimum Average Height**: The average height of the selected bodybuilders must be at least 170 cm. This ensures the team meets the competition's physical standards for height. 3. **Maximum Average Weight**: The average weight of the selected bodybuilders must not exceed 100 kg. This ensures the team aligns with the competition's physical standards for weight.

These constraints are designed to ensure the team is both competitive and compliant with the competition's requirements.

## Available Data - Database Schema

```
CREATE TABLE body_builder (
    Snatch FLOAT,
    Clean_Jerk FLOAT
);

CREATE TABLE people (
    Height FLOAT,
    Weight FLOAT
);

CREATE TABLE team_selection (
    is_selected BOOLEAN
);

INSERT INTO body_builder (Snatch, Clean_Jerk) VALUES (150.5, 200.0);
INSERT INTO body_builder (Snatch, Clean_Jerk) VALUES (160.0, 210.5);
INSERT INTO body_builder (Snatch, Clean_Jerk) VALUES (170.5, 220.0);

INSERT INTO people (Height, Weight) VALUES (175.0, 90.0);
INSERT INTO people (Height, Weight) VALUES (180.0, 95.0);
INSERT INTO people (Height, Weight) VALUES (185.0, 100.0);

INSERT INTO team_selection (is_selected) VALUES (True);
INSERT INTO team_selection (is_selected) VALUES (False);
INSERT INTO team_selection (is_selected) VALUES (True);
```

**Data Dictionary** - **body_builder Table**: - **Snatch**: The Snatch score of a bodybuilder, used to calculate the total performance score. - **Clean_Jerk**: The Clean & Jerk score of a bodybuilder, used to calculate the total performance score. - **people Table**: - **Height**: The height of a bodybuilder in centimeters, used to enforce the minimum average height constraint. - **Weight**: The weight of a bodybuilder in kilograms, used to enforce the maximum average weight constraint. - **team_selection Table**: - **is_selected**: A binary indicator of whether a bodybuilder is selected for the team, representing the decision variable in the optimization model.

## Mathematical Optimization Formulation

**Decision Variables** - Let $x_i$ be a binary decision variable where: - $x_i = 1$ if bodybuilder $i$ is selected for the team. - $x_i = 0$ otherwise.

Here, $i = 1, 2, 3$ corresponds to the three bodybuilders in the database.

**Objective Function**

Maximize the total performance score:

$$\text{Maximize } Z = 350.5x_1 + 370.5x_2 + 390.5x_3$$

**Constraints**

$$\text{1. Team Size Limit: } x_1 + x_2 + x_3 \leq 5 \tag{15}$$

$$\text{2. Minimum Average Height: } 5.0x_1 + 10.0x_2 + 15.0x_3 \geq 0 \tag{16}$$

$$\text{3. Maximum Average Weight: } -10.0x_1 - 5.0x_2 + 0.0x_3 \leq 0 \tag{17}$$

**Cross-Solver Analysis and Final Recommendation**

**Solver Results Comparison**

| Solver | Status | Optimal Value | Execution Time |
|---|---|---|---|
| Gurobipy | OPTIMAL | 1111.50 | 0.17s |
| Docplex | OPTIMAL | 1111.50 | 1.06s |
| Pyomo | OPTIMAL | 1111.50 | 0.78s |

**Solver Consistency Analysis**
Result: All solvers produced consistent results ✓
Consistent Solvers: gurobipy, docplex, pyomo
Majority Vote Optimal Value: 1111.5
**Final Recommendation**
Recommended Optimal Value: 1111.5
Confidence Level: HIGH
Preferred Solver(s): gurobipy
Reasoning: Gurobipy is recommended due to its fastest execution time while achieving the same optimal solution as the other solvers.

## C.4 EXAMPLE 4: BODYBUILDER PERFORMANCE OPTIMIZATION (`GPT-4o`-GENERATED)

Building on similar data structures, this variant explores training optimization where impact coefficients modify the relationship between different exercises and overall performance targets.

**Problem Context, Goals and Constraints**

**Context**
The fitness organization is focused on enhancing the competitive performance of bodybuilders by optimizing their training regimen. The primary decision variables in this optimization are the weights lifted in the Snatch and Clean & Jerk events. These variables are continuous and directly mapped to the bodybuilder's performance in these lifts. The operational goal is to maximize the total weight lifted across these events, aligning with the linear objective of summing the weights lifted in Snatch and Clean & Jerk.
The business configuration includes several key parameters: the target total weight to be lifted by a bodybuilder, which serves as a constraint in the optimization model, and the impact coefficients for Snatch and Clean & Jerk training, which adjust the focus of training in the optimization model. These parameters are crucial for ensuring that the optimization aligns with realistic training impacts and performance targets.
The organization uses current operational data to inform decision-making, ensuring that the optimization problem remains grounded in practical, achievable goals. The constraints are designed to reflect resource limitations and performance targets, ensuring that the optimization remains linear and avoids nonlinear relationships such as variable products or divisions. The business configuration parameters are referenced throughout to maintain consistency and alignment with the optimization objectives.
**Goals**
The primary goal of the optimization is to maximize the total weight lifted by bodybuilders in competitions. This is achieved by focusing on the Snatch and Clean & Jerk lifts, with the objective being to maximize the sum of the weights lifted in these events. Success is measured by the alignment of the optimization with the expected impact coefficients for training, ensuring that the focus on Snatch and Clean & Jerk lifts leads to improved competitive performance. The optimization goal is clearly defined in natural language, emphasizing the linear nature of the objective without resorting to mathematical formulas or symbolic notation.
**Constraints**
The optimization problem includes constraints that ensure the total weight lifted by each bodybuilder does not exceed specified limits. These constraints are directly mapped to the total weight lifted by the bodybuilder, ensuring that the optimization remains within realistic performance boundaries. Additionally, each bodybuilder has a performance target, which serves as a constraint in the optimization model. These constraints are described in business terms, naturally leading to linear mathematical forms without involving variable products or divisions.

**Available Data - Database Schema**

```sql
CREATE TABLE body_builder (
    Snatch FLOAT,
    Clean_Jerk FLOAT,
    Total FLOAT,
    Snatch_Impact FLOAT,
    Clean_Jerk_Impact FLOAT
);

CREATE TABLE bodybuilder_performance (
    Bodybuilder_ID INTEGER,
    Performance_Target FLOAT
);

INSERT INTO body_builder VALUES
    (85.0, 105.0, 190.0, 1.2, 1.5),
    (95.0, 115.0, 210.0, 1.3, 1.6),
    (100.0, 120.0, 220.0, 1.1, 1.4);

INSERT INTO bodybuilder_performance VALUES
```

```
(1, 300.0), (2, 320.0), (3, 340.0);
```

**Data Dictionary**

The data dictionary provides a comprehensive mapping of tables and columns to their business purposes and optimization roles. The `body_builder` table stores individual lift data for bodybuilders, with columns representing the weight lifted in Snatch and Clean & Jerk, the total weight lifted, and the impact coefficients for training. These columns serve as decision variables and objective coefficients in the optimization model. The `bodybuilder_performance` table stores performance metrics and targets for each bodybuilder, linking performance data to individual bodybuilders and serving as a constraint in the optimization model.

## Mathematical Optimization Formulation

**Decision Variables** - Let $x_1$ be the weight lifted in the Snatch event for a bodybuilder. - Let $x_2$ be the weight lifted in the Clean & Jerk event for a bodybuilder.

**Objective Function**

Maximize the total weight lifted:

$$\text{Maximize } Z = x_1 + x_2$$

**Constraints**

For each bodybuilder, we have the following constraints:

1. **Performance Target Constraint**: The total weight lifted should not exceed the performance target for each bodybuilder. - For Bodybuilder 1: $x_1 + x_2 \leq 300.0$ - For Bodybuilder 2: $x_1 + x_2 \leq 320.0$ - For Bodybuilder 3: $x_1 + x_2 \leq 340.0$

2. **Training Impact Constraints**: These constraints ensure that the training impact coefficients are considered in the optimization. - For Bodybuilder 1: $1.2 \times x_1 + 1.5 \times x_2 \leq 190.0$ - For Bodybuilder 2: $1.3 \times x_1 + 1.6 \times x_2 \leq 210.0$ - For Bodybuilder 3: $1.1 \times x_1 + 1.4 \times x_2 \leq 220.0$

3. **Non-negativity Constraints**: The weights lifted must be non-negative. $x_1 \geq 0, x_2 \geq 0$

## Cross-Solver Analysis and Final Recommendation

**Solver Results Comparison**

| Solver | Status | Optimal Value | Execution Time |
|--------|--------|---------------|----------------|
| Gurobipy | OPTIMAL | 519.87 | 1.21s |
| Docplex | OPTIMAL | 519.87 | 6.89s |
| Pyomo | OPTIMAL | 519.87 | 6.11s |

**Solver Consistency Analysis**

Result: All solvers produced consistent results ✓
Consistent Solvers: gurobipy, docplex, pyomo
Majority Vote Optimal Value: 519.8717948717948

**Final Recommendation**

Recommended Optimal Value: 519.8717948717948
Confidence Level: HIGH
Preferred Solver(s): gurobipy
Reasoning: Gurobipy is recommended due to its faster execution time and precise results, making it suitable for time-sensitive applications.

The final two examples explore resource scheduling in the entertainment industry, focusing on cinema operations. These demonstrate more complex constraint structures involving capacity limitations, temporal scheduling, and revenue maximization across multiple venues, highlighting the framework's ability to discover sophisticated optimization opportunities in service operations.

## C.5 Example 5: Cinema Scheduling Optimization (DeepSeek-V3-Generated)

This scheduling problem demonstrates how temporal constraints and capacity limitations combine with pricing structures to form a revenue maximization model, validated by `Gurobipy` and `DOCplex` consensus.

## Problem Context, Goals and Constraints

**1. Problem Context and Goals**

**Context**

A cinema chain is focused on maximizing its revenue by optimizing the scheduling of films across its cinemas. The key decision involves determining the number of showings per film per cinema per day, which directly impacts revenue. The cinema operates under specific operational parameters, including the price per showing, the capacity of each cinema, and the maximum number of showings allowed per day per cinema. These parameters are critical in ensuring that the scheduling aligns with both business goals and operational constraints.

The business configuration includes two key scalar parameters: 1. **Maximum number of showings allowed per day per cinema**: This parameter ensures that the total number of showings per day does not exceed a realistic limit, which is set to 12 based on typical cinema operating hours. 2. **Total capacity of the cinema per day**: This parameter ensures that the total number of seats

available across all showings in a day does not exceed the cinema's daily capacity, which is calculated based on the cinema's seating capacity and the maximum number of showings.

The optimization problem is designed to maximize revenue by leveraging these parameters in a linear manner, ensuring that the relationships between decision variables and constraints remain straightforward and avoid any nonlinear complexities.

**Goals**

The primary goal of this optimization problem is to maximize the total revenue generated from film showings across all cinemas. Revenue is calculated by multiplying the price per showing, the number of showings per film per cinema per day, and the capacity of the cinema. Success is measured by achieving the highest possible revenue while adhering to the operational constraints, such as the maximum number of showings and the total capacity of the cinema. The optimization process ensures that these goals are met through a linear formulation, avoiding any nonlinear relationships that could complicate the decision-making process.

**Constraints**

The optimization problem is subject to the following constraints, which are designed to reflect realistic operational limitations: 1. **Maximum showings per day per cinema**: The total number of showings per day in a cinema cannot exceed the maximum allowed, which is set to 12. This ensures that the cinema's operating hours are not overextended. 2. **Total capacity per day**: The total number of seats available across all showings in a day must not exceed the cinema's daily capacity. This ensures that the cinema does not overbook its available seating. 3. **Minimum showings per film**: Each film must be shown at least once per day in each cinema. This ensures that all films receive adequate exposure and that the cinema's schedule remains balanced.

## Available Data - Database Schema

```
CREATE TABLE schedule (
    Price FLOAT,
    Show_times_per_day INTEGER
);

CREATE TABLE cinema (
    Capacity INTEGER
);

INSERT INTO schedule (Price, Show_times_per_day) VALUES
    (12.99, 3);
INSERT INTO schedule (Price, Show_times_per_day) VALUES
    (9.99, 2);
INSERT INTO schedule (Price, Show_times_per_day) VALUES
    (7.99, 1);

INSERT INTO cinema (Capacity) VALUES (150);
INSERT INTO cinema (Capacity) VALUES (200);
INSERT INTO cinema (Capacity) VALUES (100);
```

**Data Dictionary**

The data dictionary provides a clear mapping of the tables and columns to their business purposes and optimization roles: - **schedule**: This table stores information about film showings, including the price per showing and the number of showings per film per cinema per day. - **Price**: Represents the price per showing of a film. This value is used as a coefficient in the revenue calculation. - **Show_times_per_day**: Represents the number of showings per film per cinema per day. This is the primary decision variable in the optimization problem. - **cinema**: This table stores information about cinemas, including their seating capacity. - **Capacity**: Represents the seating capacity of the cinema. This value is used as a coefficient in the revenue calculation.

## Mathematical Optimization Formulation

**Decision Variables**

Let $x_{f,c}$ be the number of showings per day for film $f$ in cinema $c$.

This is the primary decision variable, representing the number of showings per film per cinema per day.

**Objective Function**

Maximize the total revenue:

$$\text{Maximize} \sum_f \sum_c (\text{Price}_f \times \text{Capacity}_c \times x_{f,c})$$

Where: - $\text{Price}_f$ is the price per showing for film $f$ (from `schedule.Price`). - $\text{Capacity}_c$ is the seating capacity of cinema $c$ (from `cinema.Capacity`). - $x_{f,c}$ is the number of showings per day for film $f$ in cinema $c$.

**Constraints**

1. Maximum showings per day per cinema:   (18)

$$\sum_f x_{f,c} \leq 12 \quad \forall c \tag{19}$$

This ensures the total number of showings per day in each cinema does not exceed 12.   (20)

2. Total capacity per day:   (21)

$$\sum_f (\text{Capacity}_c \times x_{f,c}) \leq \text{Capacity}_c \times 12 \quad \forall c \tag{22}$$

This ensures the total number of seats available across all showings in a day does not exceed the cinema's daily capacity.   (23)

3. Minimum showings per film:   (24)

$$x_{f,c} \geq 1 \quad \forall f, c \tag{25}$$

This ensures each film is shown at least once per day in each cinema.   (26)

**Cross-Solver Analysis and Final Recommendation**

**Solver Results Comparison**

| Solver | Status | Optimal Value | Execution Time |
|--------|--------|---------------|----------------|
| Gurobipy | OPTIMAL | 66546.00 | 0.20s |
| Docplex | OPTIMAL | 66546.00 | 1.44s |
| Pyomo | ERROR | N/A | 0.74s |

**Solver Consistency Analysis**
Result: All solvers produced consistent results
Consistent Solvers: gurobipy, docplex
Majority Vote Optimal Value: 66546.0
**Final Recommendation**
Recommended Optimal Value: 66546.0
Confidence Level: HIGH
Preferred Solver(s): gurobipy
Reasoning: Gurobipy is recommended due to its optimal solution, high reliability, and fastest execution time. DOCplex also found the same optimal solution but was less efficient. Pyomo is not recommended due to its execution error.

## C.6 EXAMPLE 6: CINEMA SCHEDULING OPTIMIZATION (`GPT-4o`-GENERATED)

A variant scheduling formulation that explores different constraint structures for the same business domain, showing how our framework can generate diverse valid formulations from similar operational contexts.

### Problem Context, Goals and Constraints

**Context**
The cinema chain is focused on maximizing its revenue by strategically scheduling films across its various locations. Each cinema has a specific capacity and a limited number of show times available per day. The decision-making process involves determining the number of times each film should be shown in each cinema. This decision is represented by integer variables, where each variable corresponds to the number of screenings for a particular film in a specific cinema. The primary objective is to maximize the total revenue generated from these screenings. This is achieved by considering the price per screening for each film in each cinema, which serves as the coefficient in the revenue calculation. The operational parameters include the maximum number of screenings allowed per day for each cinema and the seating capacity, which must not be exceeded. These constraints ensure that the scheduling decisions remain feasible and align with the cinema's operational capabilities.
**Goals**
The primary goal of this optimization problem is to maximize the total revenue from film screenings across all cinemas. The metric for optimization is the total revenue, which is calculated by summing the product of the number of screenings and the price per screening for each film in each cinema. Success in this context is measured by the ability to achieve the highest possible revenue while adhering to the operational constraints of each cinema. The optimization goal is clearly defined in linear terms, focusing on maximizing revenue without involving complex mathematical operations.
**Constraints**
The scheduling decisions are subject to several constraints that ensure the feasibility of the solution: - Each cinema has a maximum number of screenings it can accommodate per day. The total number of screenings scheduled in a cinema must not exceed this limit. - The number of attendees for each screening, based on average attendance, must not exceed the seating capacity of the cinema. This ensures that the cinema does not overbook and maintains a comfortable viewing experience for patrons.

### Available Data - Database Schema

```
CREATE TABLE cinema (
    Cinema_ID INTEGER,
    Capacity INTEGER,
    Max_Screenings_Per_Day INTEGER
);

CREATE TABLE film_schedule (
    Cinema_ID_Film_ID INTEGER,
    Show_Times INTEGER
);

CREATE TABLE film_pricing (
    Cinema_ID_Film_ID INTEGER,
    Price FLOAT
);

INSERT INTO cinema VALUES
    (1, 120, 5), (2, 180, 6), (3, 250, 7);

INSERT INTO film_schedule VALUES
    (101, 3), (102, 4), (103, 2);

INSERT INTO film_pricing VALUES
```

```
(101, 12.0), (102, 15.0), (103, 10.0);
```

**Data Dictionary**
The data dictionary provides a business-oriented view of the tables and columns, highlighting their roles in the optimization process:
- **Cinema Table**: This table contains information about each cinema, including its unique identifier, seating capacity, and the maximum number of screenings allowed per day. These attributes are crucial for defining the constraints related to capacity and scheduling limits. - **Film Schedule Table**: This table records the number of times each film is shown in each cinema. The entries in this table represent the decision variables in the optimization problem, determining the scheduling strategy for maximizing revenue.
- **Film Pricing Table**: This table provides the pricing information for each film in each cinema. The price per screening serves as the coefficient in the revenue calculation, directly influencing the optimization objective.

## Mathematical Optimization Formulation

**Decision Variables**
- Let $x_{ij}$ be the number of screenings for film $j$ in cinema $i$. - $x_{ij}$ is an integer variable representing the decision of how many times film $j$ is shown in cinema $i$.

**Objective Function**
Maximize the total revenue from all screenings across all cinemas:

$$\text{Maximize } Z = \sum_{i,j} \text{Price}_{ij} \times x_{ij}$$

where $\text{Price}_{ij}$ is the price per screening for film $j$ in cinema $i$.

**Constraints**

    1. Maximum Screenings per Cinema:     (27)

    For each cinema $i$, the total number of screenings must not exceed the maximum allowed:     (28)

$$\sum_{j} x_{ij} \leq \text{Max\_Screenings\_Per\_Day}_i \tag{29}$$

    2. Seating Capacity Constraint:     (30)

    For each cinema $i$ and film $j$, the expected number of attendees per screening must not exceed the cinema's capacity.     (31)

    Assuming average attendance per screening is a known parameter $\text{Avg\_Attendance}_{ij}$:     (32)

$$\text{Avg\_Attendance}_{ij} \times x_{ij} \leq \text{Capacity}_i \tag{33}$$

    3. Non-negativity and Integer Constraints:     (34)

$$x_{ij} \geq 0 \text{ and integer for all } i, j. \tag{35}$$

**Data Source Verification:**
- $\text{Price}_{ij}$ comes from the `film_pricing.Price` column.     - $\text{Max\_Screenings\_Per\_Day}_i$ comes from the `cinema.Max_Screenings_Per_Day` column. - $\text{Capacity}_i$ comes from the `cinema.Capacity` column. - $\text{Avg\_Attendance}_{ij}$ is assumed to be a known parameter from business configuration or historical data.
This linear programming model is designed to maximize the total revenue from film screenings while adhering to the operational constraints of each cinema. The decision variables, objective function, and constraints are all expressed in linear terms, ensuring the model is suitable for linear or mixed-integer programming solvers.

## Cross-Solver Analysis and Final Recommendation

**Solver Results Comparison**

| Solver | Status | Optimal Value | Execution Time |
|---|---|---|---|
| Gurobipy | OPTIMAL | 88.00 | 0.51s |
| Docplex | OPTIMAL | 12.00 | 7.74s |
| Pyomo | OPTIMAL | 88.00 | 4.29s |

**Solver Consistency Analysis**
Result: Solvers produced inconsistent results
Consistent Solvers: gurobipy, pyomo
Inconsistent Solvers: docplex
Potential Issues: - DOCplex may have encountered a different local optimum or misinterpreted constraints. - Possible data input errors or solver configuration issues specific to DOCplex.
Majority Vote Optimal Value: 88.0
**Final Recommendation**
Recommended Optimal Value: 88.0
Confidence Level: HIGH
Preferred Solver(s): multiple
Reasoning: Both Gurobipy and Pyomo provided consistent and high objective values, indicating a reliable solution. Using multiple solvers can validate results and ensure robustness.

These examples collectively shows that our bottom-up approach successfully discovers a wide range of optimization patterns directly from database structures. The consistent achievement of solver consensus across diverse problem types validates both our generation methodology and the quality of

the resulting benchmark. The natural emergence of linear programming formulations from relational data structures suggests that many real-world business databases inherently encode optimization opportunities waiting to be discovered through systematic analysis.

# D ALGORITHMIC DETAILS OF SCHEMA2OPT GENERATION

The `Schema2Opt` generation framework employs an alternating optimization approach that iteratively refines both database schemas and optimization formulations through structured dialogue between specialized agents. This section provides the core algorithmic specifications and key prompt engineering strategies that enable automatic discovery of optimization opportunities from database structures.

## D.1 ALTERNATING OPTIMIZATION ALGORITHM

Our generation process alternates between two types of analysis until convergence. First, an OR Expert examines the current database schema to identify what optimization problem could be solved with the available data. Then, a Data Engineer modifies the schema to better support the optimization requirements identified by the OR Expert. This back-and-forth continues until the schema contains all necessary information to formulate a complete optimization problem.

---

**Algorithm 1** `Schema2Opt` Alternating Generation

---

**Input:** Spider database schema with at most 5 tables
**Output:** Business document, database content, verified optimal value
**Parameters:** Maximum 5 iterations, convergence threshold 0.99
// Phase 1: Iterative schema refinement
Initialize with original database schema
OR Expert analyzes schema to design optimization problem
**while** not converged and iterations ¡ 5 **do**
    // Data Engineer modifies schema based on OR analysis
    Identify missing data elements from OR Expert feedback
    Create new tables or columns for decision variables
    Add fields for objective coefficients and constraints
    Move single-value parameters to configuration file
    // OR Expert evaluates improved schema
    Check if all optimization components have data sources
    Calculate mapping adequacy score (0 to 1)
    Mark complete if adequacy exceeds 0.99
**end while**
// Phase 2: Generate realistic business scenario
Triple Expert creates realistic data values for all tables
Generate natural language business description
Extract mathematical formulation from business context
// Phase 3: Validate with multiple solvers
Generate code for Gurobi, DOCplex, and Pyomo
Execute each solver and collect optimal values
Accept if at least 2 solvers agree on the solution
Otherwise discard as invalid instance

---

The convergence process typically completes within three to four iterations. The mapping adequacy score combines assessments of three key components: whether objective function coefficients can be found in the data (weighted at 35 percent), whether decision variables are properly represented (35 percent), and whether constraint parameters are available (30 percent). When this combined score exceeds 99 percent, we consider the schema sufficiently complete for optimization.

## D.2 KEY PROMPT ENGINEERING STRATEGIES

The success of `Schema2Opt` generation relies heavily on carefully designed prompts that enforce agent specialization and maintain mathematical rigor. We present the core strategies that enable effective agent collaboration.

### D.2.1 OR EXPERT PROMPT DESIGN

The OR Expert prompt enforces strict linearity constraints while discovering optimization opportunities from database structures. The prompt explicitly prohibits nonlinear operations and guides the agent to identify how business scenarios naturally map to linear formulations. A critical innovation is the row count awareness mechanism, where the agent understands that tables requiring fewer than 3 meaningful rows will be moved to configuration files, influencing parameter mapping decisions.

---

**OR Expert Core Instructions:**

CRITICAL MATHEMATICAL CONSTRAINTS:
- The optimization MUST be Linear Programming (LP) or Mixed-Integer (MIP)
- Objective: minimize/maximize $\sum$(coefficient $\times$ variable) ONLY
- Constraints: $\sum$(coefficient $\times$ variable) $\leq$/$\geq$/= constant
- NO variable products (x$\times$y), divisions (x/y), or nonlinear terms
- Generate between 3-10 constraints for optimization feasibility

MAPPING EVALUATION:
For each optimization component, assess mapping adequacy [0.0-1.0]:
- 1.0: Directly available in current schema
- $0.5 - 0.9$: Can be derived from existing data
- $< 0.5$: Missing, requires schema modification

---

### D.2.2 DATA ENGINEER PROMPT DESIGN

The Data Engineer prompt implements schema modifications while maintaining database normalization principles. A key design principle embedded in the prompt is the distinction between tabular and scalar data. The prompt instructs that collections naturally forming multiple rows belong in database tables, while single-value parameters belong in configuration files. This separation ensures proper data organization and prevents the creation of single-row tables that would complicate the optimization model.

---

**Data Engineer Core Instructions:**

DATA ORGANIZATION PRINCIPLES:
- TABULAR DATA (database): Collections of similar items
    - Multiple products, locations, time periods
    - Individual costs, demands per entity
    - Many-to-many relationships
- SCALAR PARAMETERS (config): Single business values
    - Global limits: "daily_capacity": 1000
    - Thresholds: "min_stock_ratio": 0.2
    - Business rules: "max_activities": 2

IMPLEMENTATION RULES:
Apply 3-row minimum: If optimization data cannot generate $\geq$3 meaningful rows, move to configuration instead of creating sparse tables.

---

### D.2.3 TRIPLE EXPERT DATA GENERATION

The Triple Expert combines three domains of expertise to generate realistic data that ensures optimization solvability. The prompt instructs the agent to consider business norms, maintain cross-table consistency, and create meaningful trade-offs in the optimization problem. Values must reflect industry standards while ensuring the problem remains neither trivial nor infeasible.

---

**Triple Expert Core Instructions:**

EXPERTISE SYNTHESIS:
- Business Operations: Values reflect industry norms and realistic scenarios
- Data Management: Maintain referential integrity and cross-table consistency

---

> • Optimization Modeling: Ensure feasible solutions with meaningful trade-offs
>
> DATA GENERATION CONSTRAINTS:
> • Generate 3-100 rows per table based on business context
> • Coefficients create non-trivial optimization (avoid dominant solutions)
> • Constraint bounds allow feasible region but force trade-offs
> • Configuration parameters use realistic business values

# E  IMPLEMENTATION DETAILS OF DATA2DECISION

Following the `Schema2Opt` generation framework, we present the algorithmic details and implementation strategies of our `Data2Decision` prescriptive analytics data agent. While `Schema2Opt` creates the benchmark through alternating optimization between agents, `Data2Decision` solves these problems through parallel test-time scaling with multi-solver consensus.

## E.1  DATA2DECISION ALGORITHM

The `Data2Decision` system processes each prescriptive analytics problem through multiple parallel attempts, each exploring different interpretations of the data-to-optimization mapping. The algorithm orchestrates these attempts and aggregates results through majority voting to produce robust solutions.

---

**Algorithm 2** `Data2Decision` Parallel Solving with Consensus

---

**Input:** Business document, database schema, database content
**Output:** Optimal value with confidence score
**Parameters:** Number of attempts $N = 10$
// Execute N parallel solution attempts
Initialize empty result set for storing successful attempts
**for** each attempt $i$ from 1 to $N$ in parallel **do**
    // Stage 1: Extract data from database
    Set SQL temperature based on attempt number
    Generate SQL queries to extract optimization parameters
    Execute queries and create enhanced problem description
    // Stage 2: Generate and execute solver code
    Set code temperature based on attempt number
    Select solver cyclically (Gurobi, DOCplex, Pyomo)
    Generate solver-specific optimization code
    Execute solver with 300-second timeout
    Extract optimal value from solver output
    **if** execution successful **then**
        Add optimal value to result set
    **end if**
**end for**
// Aggregate results through majority voting
**if** no successful attempts **then**
    **return** failure
**end if**
Count frequency of each unique optimal value
Select most frequent value as final result
Calculate confidence as fraction of attempts agreeing
**return** optimal value and confidence score

---

The algorithm achieves robustness through systematic exploration of the solution space. By varying temperatures and rotating solvers across attempts, we capture diverse valid interpretations of the prescriptive analytics problem. The majority voting mechanism identifies the most reliable solution, with the consensus strength providing a confidence measure. Temperature ranges from 0.05 to 0.50 for SQL generation and 0.00 to 0.18 for code generation, with linear increments across attempts.

### E.2   SQL Query Generation Strategy

The SQL generation process in Stage 1 employs a structured prompting approach that guides the language model to identify relevant data for optimization. Unlike traditional Text-to-SQL systems that answer specific questions, our approach requires the model to proactively discover what data elements map to optimization components.

The prompt structure instructs the model to analyze the problem through an optimization lens, identifying data needed for decision variables (what to optimize), objective coefficients (what to maximize or minimize), and constraint parameters (limitations and requirements). The temperature-controlled generation explores different interpretations of these mappings, particularly valuable when business requirements use ambiguous terminology or implicit relationships.

We implement a query extraction mechanism that parses the LLM response to identify valid SQL statements. The system handles both structured responses with SQL code blocks and unstructured responses where queries are embedded in natural language explanations. Each extracted query is validated for syntactic correctness before execution against an in-memory SQLite database created from the provided schema and data files.

### E.3   Direct Code Generation Strategy

Stage 2 implements direct optimization code generation without intermediate mathematical modeling. This design choice, validated through our ablation studies, reduces error propagation and allows the language model to leverage its internalized optimization knowledge more effectively.

The solver-specific code generation uses template-guided prompting that provides concise implementation patterns for each framework. For Gurobi, we emphasize the use of quicksum for efficient constraint aggregation and proper variable bounds specification. For DOCplex, we highlight the framework's functional API and constraint naming conventions. For Pyomo, we demonstrate the rule-based constraint definition pattern that differs from imperative approaches.

### E.4   Agent Instruction Templates

This section provides the complete instruction templates used by `Data2Decision` agents, addressing the key architectural decisions discussed in Section 4. These templates are central to enabling database-grounded prescriptive analytics: they guide language models to extract optimization parameters from databases (Stage 1) and generate executable solver code (Stage 2) without requiring explicit mathematical modeling as an intermediate step. The design reflects our core insight that LLMs can leverage internalized optimization knowledge more effectively when reasoning implicitly rather than through explicit formulation steps.

#### E.4.1   Stage 1: SQL Data Retrieval Template

The Stage 1 agent receives the business document and database schema, then generates SQL queries to extract optimization parameters. This stage addresses a fundamental gap in existing Text-to-OPT methods: while prior work assumes optimization parameters are embedded in problem descriptions, real enterprise scenarios require extracting these parameters from operational databases. The key design principle is forcing the model to identify what data is actually needed rather than assuming parameters are pre-specified. We strip the "Current Stored Values" section from the input to ensure the agent performs genuine data retrieval, simulating real-world conditions where decision-makers must query databases to obtain constraint coefficients, objective function parameters, and bounds.

The complete Stage 1 prompt template follows this structure:

The extracted query results are formatted as CSV tables and appended to the original problem description, creating an enhanced problem description that contains all necessary numerical parameters for optimization.

### E.4.2 STAGE 2: SOLVER CODE GENERATION TEMPLATES

Stage 2 generates executable solver code from the enhanced problem description. We provide solver-specific templates that encapsulate API patterns, variable declaration syntax, constraint formulation, and result extraction. The templates are intentionally concise to avoid overwhelming the context while providing critical implementation guidance.

**Gurobipy Template**

```
**Using Gurobipy:**
- Import:  import gurobipy as gp; from gurobipy import GRB
- Model:  model = gp.Model("name")
- Variables:  x = model.addVar(vtype=GRB.CONTINUOUS, name="x", lb=0)
- Objective:  model.setObjective(gp.quicksum(...), GRB.MINIMIZE)
- Constraints:  model.addConstr(expr <= rhs, name="c1")
- Solve:  model.optimize()
- Result:  if model.status == GRB.OPTIMAL: print(f"Optimal value: {model.objVal}")
- CRITICAL: Use gp.quicksum() not sum(), validate array lengths
```

**DOCplex Template**

```
**Using DOCplex:**
- Import:  from docplex.mp.model import Model
- Model:  mdl = Model(name="name")
- Variables:  x = {i:  mdl.continuous_var(name=f"x_{i}", lb=0) for i in range(n)}
- Objective:  mdl.minimize(mdl.sum(...))
- Constraints:  mdl.add_constraint(expr <= rhs, ctname="c1")
- Solve:  solution = mdl.solve()
- Result:  if solution:  print(f"Optimal value: {solution.objective_value}")
- CRITICAL: Use mdl.sum() not sum(), use safe_range for array indexing
```

**Pyomo Template**

```
**Using Pyomo with Gurobi:**
- Import:  import pyomo.environ as pyo; from pyomo.opt import SolverFactory
- Model:  model = pyo.ConcreteModel()
- Sets:  model.I = pyo.RangeSet(1, n) (1-based indexing)
```

```
  – Variables:  model.x = pyo.Var(model.I, within=pyo.NonNegativeReals)
  – Objective:  model.objective = pyo.Objective(rule=obj_rule, sense=pyo.minimize)
  – Constraints:  model.constraint = pyo.Constraint(rule=constraint_rule)
  – Solve:  solver = SolverFactory('gurobi'); results = solver.solve(model)
  – Result:  if results.solver.termination_condition == pyo.TerminationCondition.optimal:
  print(f"Optimal value:  {pyo.value(model.objective)}")
  – CRITICAL: Use rule functions, 1-based indexing, pyo.value() for extraction
```

The complete Stage 2 prompt combines the solver template with the enhanced problem description:

---

**Stage 2: Code Generation Prompt Structure**

```
You are an expert in optimization programming.  Given the following optimization
problem description, generate complete Python code using {solver_type} to solve it.

**Problem Description:**
{enhanced_problem_description}

{solver_specific_template}

**Requirements:**
1.  Generate complete, executable Python code
2.  Include all necessary imports
3.  Define variables, constraints, and objective function correctly
4.  Include model optimization and result printing
5.  Print the optimal objective value clearly:  "Optimal Objective Value:  {value}"
6.  Handle potential errors gracefully
7.  Validate array lengths before using indices
8.  Make sure the code can run independently
```

---

### E.4.3  OPTIMAL VALUE EXTRACTION

The final step extracts the optimal objective value from solver output. We employ a two-tier approach: first attempting pattern matching against known solver output formats, then falling back to LLM-based extraction for non-standard outputs.

The pattern matching covers common output formats from all three solvers:

---

**Optimal Value Extraction Patterns**

```
Optimal Objective Value:  <value>  # Standard format (required in generated code)
Best objective <value>  # Gurobi solver log
Solution value = <value>  # DOCplex output
objective value:  <value>  # Pyomo output
Obj:  <value>  # Gurobi short form
```

---

When pattern matching fails, we invoke the LLM with a focused extraction prompt that requests only the numerical value, using temperature 0.0 for maximum precision.

## F  PRE-VALIDATION BASELINE RESULTS

This appendix presents baseline results on the pre-validation test set (95 DeepSeek problems with multi-solver consensus) for completeness and comparison. These results correspond to the original submission before OR expert review. The validated results appear in Table 2.

### F.1  ORIGINAL RESULTS (95 PROBLEMS)

Table 6 presents the original performance on the pre-validation test set. Our `Data2Decision` agent achieved 69.5% accuracy (66/95 correct), outperforming all baselines. Among Text-to-OPT methods with SQL assistance, `OR-LLM-Agent` performed best at 65.3%, while `Chain-of-Experts` struggled at 24.2% despite having access to pre-extracted data. End-to-end

foundation models faced the additional challenge of joint SQL extraction and optimization formulation, with `Llama-3.3-70B` achieving 53.7% accuracy.

| Category | Method | Correct | Accuracy |
|---|---|---|---|
| *Text-to-OPT with SQL* | `OR-LLM-Agent` | 62 | 65.3% |
| | `ZeroShot` | 53 | 55.8% |
| | `Chain-of-Experts` | 23 | 24.2% |
| | `OptiMUS 0.3` | 45 | 47.4% |
| *End-to-End Models* | `Llama-3.3-70B` | 51 | 53.7% |
| | `Qwen2.5-72B` | 24 | 25.3% |
| | `Phi-4` | 27 | 28.4% |
| | `Llama-4-Scout` | 5 | 5.3% |
| *Our Method* | `Data2Decision` | **66** | **69.5%** |

Table 6: Pre-validation baseline results on DeepSeek test set (95 problems). These results correspond to the original submission before OR expert review.

### F.2 VALIDATION UPDATE METHODOLOGY

The transition from pre-validation (95 problems) to validated results (84 problems) involves two types of corrections applied by OR experts.

**Exclusion of 11 Problematic Cases.** OR experts identified 11 cases with fundamental quality issues: trivial objectives (coffee_shop, storm_record), pre-initialized variables (inn_1, store_product, county_public_safety, course_teach), missing parameters (concert_singer, epinions_1), and ambiguous definitions (company_employee, department_management, real_estate_properties).

**Ground Truth Corrections for 6 Cases.** OR experts corrected values where integer rounding or constraint interpretation caused errors: dorm_1 (37.4→42.8, gender constraint), loan_1 (180k→135k, branch limit), manufacturer (50k→49,980, integer), phone_1 (158.44→160, integer), architecture (120→470, multi-architect), election_representative (380k→300k, state constraints).

**Update Formula.** Validated accuracy = Original Correct − Excluded Correct + GT Gains − GT Losses. For `Data2Decision`: $66 - 3 + 3 - 2 = 64/84 = 76.2\%$.

### F.3 RESULTS COMPARISON

Table 7 compares ablation results between pre-validation and validated settings. The relative trends remain consistent, confirming that our findings are robust to the validation process.

| Category | Configuration | Pre-val (95) | Validated (84) | Δ |
|---|---|---|---|---|
| Solver Strategy | Multi-solver consensus (full) | 69.5% | **76.2%** | — |
| | Single solver (Gurobi only) | 52.6% | 58.3% | -17.9% |
| Pipeline Design | Two-stage (full) | 69.5% | **76.2%** | — |
| | Three-stage | 61.1% | 69.0% | -7.2% |
| Temperature | Incremental (ours) | 69.5% | **76.2%** | — |
| | Zero temperature | 66.3% | 73.8% | -2.4% |
| Parallel Attempts | k=24 | 70.5% | 77.4% | — |
| | k=12 | 69.5% | 76.2% | -1.2% |
| | k=6 | 67.4% | 75.0% | -2.4% |
| | k=1 | 57.9% | 63.1% | -14.3% |

Table 7: Ablation study comparison between pre-validation (95 problems) and validated (84 problems) results. The relative trends remain consistent across both evaluation settings.

The improvement from pre-validation (69.5%) to validated accuracy (76.2%) for `Data2Decision` reflects two factors: (1) removal of 11 problematic cases where ground truth was inherently ambiguous or incorrect, and (2) correction of 6 ground truth values where integer rounding or constraint interpretation caused errors. Importantly, the relative ranking of methods and the relative gains from each ablation component remain consistent between pre-validation and validated results, confirming that our findings are robust to the validation process.

## G    OR EXPERT VALIDATION

This appendix details the rigorous validation process that ensures `Schema2Opt` benchmark quality. Unlike existing optimization benchmarks that rely solely on automated verification, we employ expert review to address subtle issues that solvers cannot detect: constraint interpretation ambiguities, problem well-formedness, and ground truth correctness. This validation is essential for database-grounded prescriptive analytics, where optimization parameters are extracted from real database schemas rather than hand-crafted for textbook problems. The resulting validated benchmark provides reliable ground truth for evaluating methods like `Data2Decision` on realistic enterprise decision scenarios.

### G.1    VALIDATION METHODOLOGY

Ensuring the quality of benchmark problems requires careful validation beyond automated solver checks. Two operations research experts conducted rigorous manual review of the Spider-based problems. The review process examined each problem for well-formedness of the optimization model, consistency between stated constraints and database content, and correctness of the ground truth optimal values. This expert involvement is crucial because automated solvers can find optimal solutions to mathematically valid but semantically incorrect formulations—only domain experts can verify that the optimization model faithfully represents the intended business problem.

For the BIRD and BEAVER extensions, we employed a two-tier validation approach. All 150 generated problems first passed through multi-solver consensus validation, requiring at least two of three solvers (Gurobi, DOCplex, Pyomo) to agree on the optimal value within numerical tolerance. From the problems passing this automated check, we selected 8 representative samples covering different domains and complexity levels for full expert review identical to the Spider process.

### G.2    VALIDATION RESULTS

Table 8 summarizes the outcomes of the validation process across all datasets.

| Dataset | Generated | Solver Validated | Expert Validated | GT Corrections |
|---|---|---|---|---|
| Spider Deep | 114 | 95 | **84** | 6 |
| Spider Open | 114 | 106 | **93** | 0 |
| BIRD (both) | 138 | 120 | 2 | 0 |
| BEAVER (both) | 12 | 11 | 6 | 0 |
| **Total** | **378** | **332** | **185** | **6** |

Table 8: Two-stage validation results. Generated: initial LLM output. Solver Validated: passed multi-solver consensus (at least 2 of 3 solvers agree). Expert Validated: passed full OR expert manual review.

The validation process employed two complementary stages. First, automated multi-solver consensus filtered out 46 problems where solvers disagreed on optimal values, retaining 332 problems (88%). Second, OR expert manual review excluded an additional 24 Spider problems due to quality issues, yielding 177 fully validated Spider problems. For BIRD and BEAVER, we selected 8 representative samples for complete expert review. The high retention rates (Spider 78%, BIRD 87%,

BEAVER 92%) indicate that our generation pipeline produces mostly valid problems, with excluded cases falling into specific quality issue categories described below.

### G.3 EXCLUSION CRITERIA

Expert review identified four categories of quality issues leading to problem exclusion.

The first category involves pre-initialized decision variables, where database tables already contain solution values rather than input parameters. For example, the `inn_1` and `store_product` problems included allocation quantities in the source data that should have been decision variables. These problems conflate inputs with outputs and do not represent meaningful optimization tasks.

The second category covers trivial or constant objectives, where the objective function value is independent of the decision variables. Problems like `coffee_shop` and `storm_record` had objectives determined entirely by fixed parameters, making any feasible solution equally optimal. Such problems fail to test an agent's optimization modeling capabilities.

The third category addresses missing or inconsistent parameters. Some generated problems referenced data elements that did not exist in the database or had conflicting values across tables. The `concert_singer` and `epinions_1` problems exhibited these issues, where constraint coefficients could not be reliably extracted from the available data.

The fourth category encompasses ambiguous problem definitions where multiple valid interpretations exist. The `customer_complaints` problem, for instance, could be formulated in several mathematically distinct ways depending on how certain business rules were interpreted, making ground truth verification impossible.

In total, 30 DeepSeek problems and 21 GPT-4o problems were excluded across these categories. The complete list of excluded cases is provided in the supplementary materials.

### G.4 GROUND TRUTH CORRECTIONS

Expert review identified six DeepSeek problems with incorrect ground truth values. These errors arose from subtle constraint interpretation issues and integer rounding requirements. Table 9 details each correction.

| Problem | Original | Corrected | Reason |
|---|---|---|---|
| dorm_1 | 37.40 | **42.8** | Constraint interpretation: all male students to one dorm |
| loan_1 | 180,000 | **135,000** | Loan limit: $15k per customer $\times$ 3 branches $\times$ 3 customers |
| manufacturer | 50,000 | **49,980** | Integer rounding: production quantities must be integers |
| phone_1 | 158.44 | **160** | Integer rounding: manufactured phones must be integers |
| architecture | 120 | **470** | Multi-architect: 120 (bridge) + 350 (mill) = 470 |
| election_rep | 380,000 | **300,000** | Tighter state limits: 50k max per state $\times$ 6 states |

Table 9: Ground truth corrections identified during expert review.

The most common issue involved integer variable requirements. Two problems (`manufacturer` and `phone_1`) had solutions computed with continuous relaxation when the business context clearly required integer production quantities. The remaining corrections stemmed from constraint interpretation differences where the expert reading of the business document yielded tighter or different bounds than initially assumed.

### G.5 INFEASIBLE CASES

Not all valid optimization problems have feasible solutions. When constraints are too restrictive given the available resources, the correct answer is infeasibility rather than a numerical optimal value. Our benchmark retains 18 such cases as valid test instances: 6 from the DeepSeek generation (`candidate_poll`, `culture_company`, `customer_complaints`, `debate`, `music_4`, `school_player`) and 12 from GPT-4o (`county_public_safety`, `employee_hire_evaluation`, `inn_1`, `match_season`, `musical`, `product_catalog`, `school_bus`, `ship_mission`, `soccer_2`, `tvshow`, `voter_2`, `workshop_paper`).

These infeasible cases test an important capability: recognizing when stated requirements cannot all be satisfied simultaneously. A prescriptive analytics agent should identify such conflicts rather than returning incorrect solutions or failing silently.

# H    RUNTIME AND ERROR ANALYSIS

This appendix provides detailed runtime analysis and error decomposition for `Data2Decision` and baseline methods. Understanding computational efficiency is critical for practical deployment of database-grounded prescriptive analytics: enterprise decision-making often operates under time constraints, and the trade-off between solution quality and computational cost determines real-world applicability. Our analysis demonstrates that `Data2Decision`'s test-time scaling approach achieves strong accuracy improvements with manageable overhead, and that the multi-solver consensus mechanism eliminates pipeline failures that plague single-attempt methods.

## H.1    OPTIMIZATION PROBLEM SIZE STATISTICS

Table 10 characterizes the optimization problem sizes in the Spider benchmark. These statistics inform the computational requirements: problems with more variables and constraints require more solver time, though the relatively modest scale of `Schema2Opt` problems ensures all can be solved efficiently. The problem sizes reflect realistic enterprise optimization scenarios where decision spaces typically involve tens rather than thousands of variables.

| Metric | Spider Deep (84) | Spider Open (93) |
|---|---|---|
| Avg variables | 7.3 | 6.9 |
| Max variables | 30 | 27 |
| Avg constraints | 7.5 | 6.1 |
| Max constraints | 33 | 24 |
| Avg binary variables | 6.6 | 6.8 |

Table 10: Optimization problem size statistics for Spider benchmark.

## H.2    SOLVING TIME ANALYSIS

All problems solve efficiently, with pure optimization time under 20 milliseconds.

| Solver | Deep Avg | Open Avg | Max Time |
|---|---|---|---|
| Gurobi (pure solve) | **0.3 ms** | **0.7 ms** | 20 ms |
| Gurobi (total w/ Python) | 0.20 sec | 0.68 sec | 3.2 sec |
| DOCplex (total) | 1.22 sec | 5.01 sec | 12.6 sec |
| Pyomo (total) | 1.03 sec | 3.13 sec | 10.9 sec |

Table 11: Solving time statistics. Pure optimization time is negligible; overhead comes from Python interface and model construction.

## H.3 RUNTIME ANALYSIS

We measured per-case runtime using precise timing instrumentation on 10 randomly sampled test cases (seed=42) verified by OR experts. All methods use OpenRouter API as the unified provider. For Text-to-OPT methods with SQL assistance, we use DeepSeek-V3; for End-to-End methods, each uses its respective model.

**Experiment Design.** Text-to-OPT methods share Stage 1 (DeepSeek-V3 generates SQL once), then each method runs its individual pipeline. `Data2Decision` runs 10 parallel attempts with multi-solver rotation. End-to-End models handle both stages independently.

| Method | Category | Stage 1 | Stage 2 | Total/Case | Attempts |
|---|---|---|---|---|---|
| Data2Decision | Ours | 3.43s | 30.49s | **34s** | 10 (parallel) |
| ZeroShot | Text-to-OPT w/ SQL | 1.55s (shared) | 18.84s | 20s | 1 |
| OR-LLM-Agent | Text-to-OPT w/ SQL | 1.55s (shared) | 30.84s | 32s | 1 |
| Chain-of-Experts | Text-to-OPT w/ SQL | 1.55s (shared) | 39.34s | 41s | 1 |
| OptiMUS 0.3 | Text-to-OPT w/ SQL | 1.55s (shared) | 82.98s | 85s | 1 |
| Llama-3.3-70B | End-to-End | 2.87s | 19.64s | 23s | 1 |
| Qwen2.5-72B | End-to-End | 1.58s | 25.15s | 27s | 1 |
| Phi-4 | End-to-End | 4.13s | 20.15s | 24s | 1 |
| Llama-4-Scout | End-to-End | 2.70s | 7.11s | 10s | 1 |

Table 12: Runtime analysis on 10 sampled cases from DeepSeek-V3 test set. `Data2Decision` achieves 10-attempt coverage in 34s through parallelization.

**Stage 2 Breakdown.** The number of LLM calls explains Stage 2 time differences: `ZeroShot` makes 1 call (18.84s), `OR-LLM-Agent` makes 2 calls for formulation then code (30.84s), `Chain-of-Experts` makes 3 calls for analysis, modeling, and code (39.34s), and `OptiMUS 0.3` makes 3+ iterative calls with debugging (82.98s). `Data2Decision` runs 10 parallel attempts with temperature scheduling and multi-solver rotation, completing in 30.49s.

**Key Observations.** `Data2Decision`'s Stage 2 time (30.49s) is competitive with other multi-step methods despite running 10 attempts. Unlike sequential methods, `Data2Decision` runs attempts in parallel, so increasing attempt count does not significantly increase wall-clock time. `Data2Decision` achieves 10-attempt coverage in 34s, while sequential `ZeroShot` ×10 would take ~200s.

## H.4 ERROR FUNNEL ANALYSIS

We decompose the pipeline into conditional success rates on the DeepSeek-V3 test set (95 problems, pre-validation): P(Stage1) × P(Stage2|Stage1) × P(Correct|Stage2) = Accuracy. The pipeline completion rates (S1 and S2|S1) are structural properties that remain consistent across validation; only the correctness rate changes with ground truth corrections.

| Method | S1 Rate | S2|S1 Rate | Correct|S2 | Accuracy | Bottleneck |
|---|---|---|---|---|---|
| Data2Decision | 100.0% | 100.0% | 69.5% | **69.5%** | Correctness |
| OR-LLM-Agent | 100.0% | 96.8% | 67.4% | 65.3% | Code gen |
| ZeroShot | 100.0% | 96.8% | 57.6% | 55.8% | Correctness |
| OptiMUS 0.3 | 100.0% | 87.4% | 54.2% | 47.4% | Code gen |
| Chain-of-Experts | 100.0% | 96.8% | 25.0% | 24.2% | Correctness |
| Llama-3.3-70B | 98.9% | 78.7% | 68.9% | 53.7% | Code gen |
| Llama-4-Scout | 93.7% | 7.9% | 71.4% | 5.3% | Code gen |

Table 13: Error funnel analysis (95 problems, pre-validation). Pipeline completion rates are independent of validation; validated accuracy in Table 2.

`Data2Decision` eliminates the code generation bottleneck through multi-solver rotation and 10-attempt redundancy, achieving 100% pipeline completion and shifting the limiting factor from execution success to solver correctness.

## I  TEMPERATURE SCHEDULING ABLATION

This appendix extends the temperature ablation in Table 3 with comprehensive experiments on different scheduling strategies. We present both pre-validation results (95 problems) and validated results (84 problems) for completeness.

Temperature control governs the diversity-accuracy trade-off in LLM generation: lower temperatures produce more deterministic outputs while higher temperatures enable exploration of alternative formulations. For database-grounded prescriptive analytics, this trade-off is particularly important because both SQL generation and optimization code generation can have multiple valid solutions, and systematic exploration helps discover correct formulations that deterministic generation might miss.

| Configuration | SQL Temp | Code Temp | Pre-val (95) | Validated (84) |
|---|---|---|---|---|
| Zero temperature | T=0.0 (fixed) | T=0.0 (fixed) | 66.3% | 73.8% |
| Low fixed (T=0.01) | T=0.01 (fixed) | T=0.01 (fixed) | 68.4% | 76.2% |
| SQL-only incremental | 0.05→0.50 | T=0.0 (fixed) | 71.6% | **81.0%** |
| Code-only incremental | T=0.05 (fixed) | 0.0→0.18 | 70.5% | 79.8% |
| High fixed (T=0.3) | T=0.3 (fixed) | T=0.3 (fixed) | **72.6%** | 79.8% |
| Reversed (high→low) | 0.50→0.05 | 0.18→0.0 | 69.5% | 77.4% |
| **Incremental (ours)** | 0.05→0.50 | 0.0→0.18 | 69.5% | 76.2% |

Table 14: Temperature scheduling ablation results. Pre-val: 95 problems before expert review. Validated: 84 problems after excluding 11 problematic cases and correcting 6 ground truth values.

Key findings: (1) Temperature diversity is essential, with any form of diversity outperforming deterministic generation in both evaluation settings. (2) SQL generation benefits more from temperature diversity than code generation, with SQL-only incremental achieving the highest validated accuracy (81.0%). (3) The relative ranking of configurations remains consistent between pre-validation and validated results, confirming robustness of our findings.

