# OpenReview forum: "Data2Decision: A Prescriptive Analytics Data Agent Bridging Enterprise Information and Optimal Decisions"
_ICLR.cc/2026/Conference — ICLR 2026 Conference Desk Rejected Submission_

### Official Review · Reviewer_iACw · 2025-10-27

**Soundness:** 2
**Presentation:** 2
**Contribution:** 3
**Rating:** 4
**Confidence:** 4

**Summary:**

This paper formulates a prescriptive analytic problem in real-world enterprise decision-making scenarios, which requires extracting optimization parameters from enterprise databases. Based on the problem formulation, this paper presents Schema2Opt, a framework for synthetic dataset generation that discovers optimization problems from database schemas. This paper further introduces a Data2Decision framework for database-grounded prescriptive analytics that first extracts optimization parameters from databases and then solves the optimization problem.

**Strengths:**

The paper formulates a new prescriptive analytic problem based on real-world enterprise applications with the shift to a bottom-up paradigm. The proposed Schema2Opt dataset generation framework can address the lack of realistic benchmark for database-grounded prescriptive analytics. The paper introduces the first agentic framework Data2Decision to solve database-grounded prescriptive analytic problems. The technical details of Schema2Opt are well presented.

**Weaknesses:**

While the motivation of the paper is clear and Schema2Opt is well presented, the presentation of Data2Decision lacks details, such as the instruction templates for agents. The techniques proposed for Data2Decision seem incremental, such as the one in Section 4.3. It is unclear how some of the techniques would benefit databased-grounded prescriptive analytics.
The experimental setting in Section 5.1 is not detailed enough. The information on the backbone model for Data2Decision is not provided. Further information on baselines such as ZeroShot would be appreciated.
Experimental evaluation is not thorough. Evaluations on some real-world datasets would be helpful for a fairer comparison. The parameter analysis is incomplete. The authors Could perform analysis on adaptive temperature scheduling. It would also be interesting to see which of the two stages in Data2Decision contributes more to the performance gains over baselines.
I assume the backbone models for Data2Decision are the same as those in Schema2Opt, i.e., GPT-4o and DeepSeek-V3. Would this introduce information leakage and offer extra advantages to Data2Decision? Another concern I have is that most of the problems generated by Schema2Opt are simple ones according to Figure 4, which hampers the usefulness of the benchmark.

**Questions:**

What is the backbone model of Data2Decision in the experiments?
Could the authors please confirm whether it is GPT-4o, GPT-4o-mini, or both that are used in the experiments?
Would it be reasonable to also evaluate GPT-4o and DeepSeek-V3 in the end-to-end models?
Typos, such as the sentence repetition in lines 303-305.

---

> ### Author Response · Authors · 2025-11-28
>
> We thank Reviewer iACw for the constructive feedback. Your suggestions improved our clarity. We added (1) instruction templates, (2) stage-wise analysis, (3) temperature ablation, (4) backbone model details. Responses below.
>
> > Q1. "**While the motivation of the paper is clear and Schema2Opt is well presented, the presentation of Data2Decision lacks details, such as the instruction templates for agents.**"
>
> **A1.** We thank the reviewer for this feedback. We have added complete agent instruction templates in **Appendix E.4**. The key templates include:
>
> **Stage 1 (SQL Generation):** The agent receives problem context and database schema, then generates SQL queries to extract optimization parameters. The complete template is provided in Appendix E.4.
>
> ```
> You are an expert database analyst helping with optimization problem data retrieval.
> Based on the problem description and database schema, analyze what data would be
> most useful for solving this optimization problem...
>
> **Analysis Guidelines:**
> - Identify what data is needed for decision variables (what needs to be optimized)
> - Identify what data is needed for objective function coefficients
> - Identify what data is needed for constraint parameters
> - Consider relationships between tables and potential joins
> ...
> ```
>
> **Stage 2 (Code Generation):** We provide solver-specific templates for Gurobipy, DOCplex, and Pyomo that encapsulate API patterns, variable declaration, constraint syntax, and result extraction. For example, the Gurobipy template includes:
>
> ```
> Using Gurobipy:
> Import: import gurobipy as gp; from gurobipy import GRB
> Model: model = gp.Model("name")
> Variables: x = model.addVar(vtype=GRB.CONTINUOUS, name="x", lb=0)
> Objective: model.setObjective(gp.quicksum(...), GRB.MINIMIZE)
> Constraints: model.addConstr(expr <= rhs, name="c1")
> CRITICAL: Use gp.quicksum() not sum(), validate array lengths
> ...
> ```
>
> Similar templates are provided for DOCplex and Pyomo with their respective API conventions. The optimal value extraction patterns are also documented in Appendix E.4.
>
>
> ---
>
> > Q2. "**The techniques proposed for Data2Decision seem incremental, such as the one in Section 4.3.**"
>
> **A2.** We appreciate the concern. Section 4 may appear incremental in isolation, but its components are essential for solving a new task: database-grounded prescriptive analytics, where the agent must perform the full pipeline (database tables → SQL → parameters → MILP → solver) without pre-embedded optimization data. Existing Text-to-OPT systems assume parameters are given; none operate in this setting. Importantly, our cross-language formulation agreement—which leverages the equivalence of optimization models across multiple solver libraries as a self-consistency test—is a novel mechanism in this domain and significantly improves performance.
>
> The Section 4 methods provide non-trivial gains on the validated 84-problem DeepSeek test set:
> direct code generation avoids a modeling step that drops accuracy from 76.2%→69.0% (Table 3);
> multi-solver-language diversification yields the largest lift (76.2%→58.3% without it);
> test-time scaling and temperature scheduling address SQL and modeling uncertainty.
>
> Thus, the contributions are not incremental but necessary innovations for the first end-to-end solution to this new problem.

---

> ### Author Response · Authors · 2025-11-28
>
> > Q3. "**It is unclear how some of the techniques would benefit database-grounded prescriptive analytics.**"
>
> **A3.** (1) Two-stage pipeline. Our task requires both data extraction and optimization modeling. These are fundamentally different capabilities. Stage 1 (SQL generation) identifies which tables and columns contain optimization parameters. Stage 2 (code generation) formulates the mathematical model. This modular design decouples data preparation from optimization modeling, allowing each stage to be optimized independently. It also enables future extensions such as data-agnostic optimization models that treat parameters as inputs rather than hardcoded values.
> (2) Multi-solver-language diversification. Unlike traditional Text-to-OPT where the problem is fully specified, database-grounded settings involve ambiguity in how schema elements map to optimization components. Different solver languages (Gurobi, DOCplex, Pyomo) encode different modeling patterns from their documentation and training data. Cycling through solvers explores diverse valid interpretations of the same business problem. This yields the largest performance gain (76.2%→58.3% without it on validated data).
> (3) Test-time scaling and temperature scheduling. Database-grounded problems have uncertainty at both stages. SQL queries may extract different subsets of relevant data. The same data may admit multiple valid formulations. Sampling multiple attempts with varying temperatures explores this solution space. Majority voting identifies the most robust interpretation.
> (4) Direct code generation. Skipping explicit mathematical formulation reduces error propagation in a multi-stage pipeline. This is especially important when parameters come from database queries rather than clean problem descriptions.
> These techniques are not generic improvements but targeted solutions to challenges unique to our setting.
>
> ---
>
> > Q4. "**The experimental setting in Section 5.1 is not detailed enough. The information on the backbone model for Data2Decision is not provided. Further information on baselines such as ZeroShot would be appreciated.**"
>
> **A4.** We thank the reviewer for this question. Section 5.1 specifies the backbone models where we state that Text-to-OPT baselines use "the corresponding backbone model (GPT-4o-mini for the GPT-4o test set, DeepSeek-V3 for the DeepSeek-V3 test set)" for the unified first stage. Data2Decision uses the same backbone configuration. We have made this clearer in the revised Section 5.1. For additional clarity, ZeroShot performs single-attempt direct code generation without intermediate modeling steps. All Text-to-OPT baselines share identical Stage 1 outputs and differ only in their Stage 2 formulation methods. End-to-end models use their own backbone for both stages.
>
>
> ---
>
> > Q5. "**Experimental evaluation is not thorough. Evaluations on some real-world datasets would be helpful for a fairer comparison.**"
>
>
> **A5.** We agree that real-world evaluations are valuable. However, releasing real enterprise datasets for prescriptive decision-making is extremely challenging due to data privacy, confidentiality, and IP restrictions. Real operational databases contain sensitive business rules, financial constraints, and proprietary optimization logic that organizations cannot share publicly. This challenge is well documented in the prescriptive analytics literature and is a primary reason why existing Text-to-OPT benchmarks (e.g., NL4Opt, NLP4LP, ReSocratic) rely on synthetic or heavily simplified examples rather than real enterprise data.
>
> To address this limitation, our work introduces Schema2Opt, a bottom-up synthetic framework designed specifically to approximate real-world enterprise workflows while preserving privacy. The generation pipeline ensures realistic schemas, meaningful optimization structure, and ground truth verified through multi-solver-language consensus. This makes the benchmark suitable for fair comparison across methods without exposing sensitive business information.
>
> Additionally, we have extended Schema2Opt to BIRD and BEAVER real-world databases as documented in **Section 3.6** and **Appendix B.2 through B.3**.

---

> ### Author Response · Authors · 2025-11-28
>
> ---
>
> > Q6. "**The parameter analysis is incomplete. The authors could perform analysis on adaptive temperature scheduling.**"
>
>
> **A6.** We ran temperature ablation on both pre-validation (95 cases) and validated (84 cases) DeepSeek test sets with 10 parallel attempts each. We tested 7 configurations varying temperature scheduling strategy. This analysis is now documented in **Appendix I** with **Table 14**.
>
> | Configuration | SQL Temp | Code Temp | Pre-val (95) | Validated (84) |
> |---------------|----------|-----------|--------------|----------------|
> | Zero temperature | 0.0 fixed | 0.0 fixed | 66.3% | 73.8% |
> | Low fixed | 0.01 fixed | 0.01 fixed | 68.4% | 76.2% |
> | SQL-only incremental | 0.05→0.50 | 0.0 fixed | 71.6% | 81.0% |
> | Code-only incremental | 0.05 fixed | 0.0→0.18 | 70.5% | 79.8% |
> | High fixed | 0.3 fixed | 0.3 fixed | 72.6% | 79.8% |
> | Reversed | 0.50→0.05 | 0.18→0.0 | 69.5% | 77.4% |
> | **Incremental (ours)** | 0.05→0.50 | 0.0→0.18 | 69.5% | 76.2% |
>
> We found that (1) diversity matters more than the specific scheduling pattern. Zero temperature achieves 66.3%/73.8%, while any diversity strategy reaches 69.5% to 72.6% pre-validation. (2) SQL generation benefits more from diversity than code generation, with SQL-only incremental reaching 71.6%/81.0% compared to 70.5%/79.8% for code-only. This indicates that data extraction uncertainty dominates optimization formulation uncertainty. (3) The direction of temperature change does not matter since reversed and incremental perform similarly.
>
> We chose incremental scheduling for its practical advantages. Low temperature in early attempts ensures high-quality baselines and enables early stopping when consensus is reached quickly. High temperature in later attempts provides systematic exploration only when needed.
>
>
> ---
>
> > Q7. "**It would also be interesting to see which of the two stages in Data2Decision contributes more to the performance gains over baselines.**"
>
> **A7.**  We analyze stage contributions through error funnel decomposition on the DeepSeek pre-validation test set (95 cases). Pipeline completion rates are structural properties independent of validation. This analysis is documented in **Appendix H.4** with **Table 13**.
>
> Database-grounded prescriptive analytics requires two sequential capabilities: (a) extracting optimization parameters from databases via SQL, and (b) formulating and solving optimization models via code generation. We decompose the pipeline into conditional success rates: S1 Rate measures Stage 1 completion, S2|S1 Rate measures Stage 2 completion given Stage 1 success, and Correct|S2 Rate measures solver correctness given Stage 2 success. Final accuracy equals S1 × S2|S1 × Correct|S2.
>
> | Method | S1 Rate | S2\|S1 Rate | Correct\|S2 | Accuracy | Bottleneck |
> |--------|---------|-------------|-------------|----------|------------|
> | Data2Decision | 100.0% | 100.0% | 69.5% | **69.5%** | Solver correctness |
> | OR-LLM-Agent | 100.0% | 96.8% | 67.4% | 65.3% | Code generation |
> | ZeroShot | 100.0% | 96.8% | 57.6% | 55.8% | Solver correctness |
> | OptiMUS 0.3 | 100.0% | 87.4% | 54.2% | 47.4% | Code generation |
> | Chain-of-Experts | 100.0% | 96.8% | 25.0% | 24.2% | Solver correctness |
> | Llama-3.3-70B | 98.9% | 78.7% | 68.9% | 53.7% | Code generation |
> | Llama-4-Scout | 93.7% | 7.9% | 71.4% | 5.3% | Code generation |
>
> Note: Accuracy column shows pre-validation results. Validated accuracy on 84 problems (76.2% for Data2Decision) appears in Table 2.
>
> **(1) Stage 1 (SQL extraction) is not the bottleneck.** All methods achieve S1 Rate ≥93.7%. Text-to-OPT methods with SQL assistance use shared pre-computed Stage 1 (100%), while end-to-end foundation models independently achieve 93.7% to 98.9%. SQL generation and database parameter extraction are relatively well-solved. **(2) Stage 2 (code generation) is the critical differentiator.** S2|S1 rates vary from 7.9% (Llama-4-Scout) to 100% (Data2Decision). Most baselines fail at code generation due to syntax errors or API misuse. Llama-4-Scout shows this clearly: only 7.9% of its code executes, but when it does, 71.4% are correct. Data2Decision eliminates this bottleneck through multi-solver-language consensus (+17.9% on validated data), 10-attempt redundancy, and two-stage pipeline (+7.2%). These achieve full pipeline completion, shifting the bottleneck to solver correctness.

---

> ### Author Response · Authors · 2025-11-28
>
> > Q8. "**I assume the backbone models for Data2Decision are the same as those in Schema2Opt, i.e., GPT-4o and DeepSeek-V3. Would this introduce information leakage and offer extra advantages to Data2Decision?**"
>
>
> **A8.** We thank the reviewer for this question and clarify there is no information leakage. Schema2Opt uses GPT-4o and DeepSeek-V3 to generate problems, while Data2Decision uses GPT-4o-mini and DeepSeek-V3 to solve them. Note that for the GPT test set, we use a different and smaller model (GPT-4o-mini). All Text-to-OPT baselines use the identical backbone models as Data2Decision for fair comparison, as stated in Section 5.1. Problem generation and problem solving are fundamentally different tasks. Model weights contain no problem-specific knowledge, only general optimization and SQL reasoning abilities. If leakage existed, we would expect better performance on same-model test sets, but the pattern shows otherwise: DeepSeek-V3 achieves 76.2% while GPT-4o-mini achieves only 59.1%, suggesting different problem characteristics rather than leakage. We have clarified this setup more explicitly in the revised Section 5.1.
>
> ---
>
> > Q9. "**Another concern I have is that most of the problems generated by Schema2Opt are simple ones according to Figure 4, which hampers the usefulness of the benchmark.**"
>
> **A9.** We thank the reviewer for raising this important concern. We address it from three perspectives.
>
> First, our goal is not to mirror enterprise-scale optimization models. We aim to evaluate whether LLMs can perform the entire multi-step data-to-decision pipeline: database tables → SQL queries → parameter extraction → optimization modeling → optimal decisions. This end-to-end capability is the core contribution, and simpler problems serve as a meaningful starting point for this novel task.
>
> Second, we extended Schema2Opt to BIRD [1] and BEAVER [2] datasets, generating more complex problems. BIRD contains large-scale databases from 37 professional domains. BEAVER uses MIT's actual operational systems. We generated 131 valid problems with ≥2 solver consensus. Due to rebuttal time constraints, we fully validated 8 samples (2 from BIRD, 6 from BEAVER) with OR expert verification and include them in the supplementary materials. We will release additional validated cases upon acceptance. This extension is documented in **Section 3.6** and **Appendix B.2 through B.3**.
>
> | Metric | BIRD (DeepSeek-V3) | BIRD (GPT-4o) | BEAVER (DeepSeek-V3) | BEAVER (GPT-4o) |
> |--------|--------------------|--------------------|----------------------|----------------------|
> | Problems generated | 69 | 69 | 6 | 6 |
> | ≥2 solver consensus | 57 | 63 | 6 | 5 |
> | Avg tables/problem | 4.7 | 3.3 | 4.3 | 3.7 |
> | Max tables/problem | 10 | 6 | 7 | 6 |
> | Avg rows/table | 7.0 | 8.0 | 6.7 | 7.6 |
>
> These extensions demonstrate that our framework handles larger and more realistic schemas.
>
> Third, in many MILP formulations, the core model structure (variable types, objective patterns, constraint logic) is largely invariant to problem scale. The code required to express these models is similarly scale-invariant since both small and large instances use arrays and indexed sets. Smaller instances still capture the modeling and reasoning challenges that matter for evaluating LLM capabilities. This observation aligns with existing optimization modeling benchmarks like NL4Opt and ComplexOR, which also contain mostly small-scale problems.
>
> We appreciate the reviewer's feedback. **Appendix B.4** (Cross-Dataset Comparison) demonstrates that our generation approach scales across schema complexity levels while maintaining solver tractability.
>
> **References**
>
> [1] Li et al. "Can LLM already serve as a database interface? A big bench for large-scale database grounded text-to-SQLs." NeurIPS 2024.
>
> [2] Chen et al. "BEAVER: An enterprise benchmark for text-to-SQL." arXiv:2409.02038 (2024).

---

> ### Author Response · Authors · 2025-11-28
>
> > Q10. "**What is the backbone model of Data2Decision in the experiments? Could the authors please confirm whether it is GPT-4o, GPT-4o-mini, or both that are used in the experiments?**"
>
> **A10.** We thank the reviewer for this clarification request. Data2Decision uses test-set-specific backbone models: GPT-4o-mini for the GPT-4o test set and DeepSeek-V3 for the DeepSeek-V3 test set. Both models are used for SQL generation (Stage 1) and code generation (Stage 2). This is now clarified in **Section 5.1**.
>
> ---
>
> > Q11. "**Would it be reasonable to also evaluate GPT-4o and DeepSeek-V3 in the end-to-end models?**"
>
> **A11.** We thank the reviewer for this suggestion. Our ZeroShot baseline already evaluates DeepSeek-V3 as an end-to-end model. It uses DeepSeek-V3 for both Stage 1 (SQL generation) and Stage 2 (code generation) with a single attempt. For the GPT-4o test set, we use GPT-4o-mini for both stages.
>
>
> ---
>
> > Q12. "**Typos, such as the sentence repetition in lines 303-305.**"
>
> **A12.** We thank the reviewer for catching this error. We have corrected the sentence repetition in the revised version.
>
>
> ---
>
> Thank you for the helpful comments. We have incorporated all clarifications in the revision. Feel free to let us know any follow-up questions! We are happy to discuss any remaining ones.

---

### Official Review · Reviewer_vUcr · 2025-10-30

**Soundness:** 4
**Presentation:** 2
**Contribution:** 3
**Rating:** 4
**Confidence:** 5

**Summary:**

This paper presents Data2Decision, an LLM-based “prescriptive analytics agent” that supposedly bridges database querying and optimization modeling. The system works in two stages:

It generates SQL queries to extract parameters (decision variables, coefficients, constraints) from relational databases.

It converts the extracted content into solver code (e.g., Gurobi, Pyomo) to find optimal decisions.

To evaluate this, the authors propose Schema2Opt, a synthetic benchmark created by converting SQL schemas (from the Spider dataset) into optimization problems via alternating LLM agents (“OR Expert” and “Data Engineer”). The authors claim this is the first bottom-up benchmark for prescriptive analytics, arguing that prior Text-to-OPT work (e.g., ORLM, OptiMUS, Chain-of-Experts) used oversimplified textbook problems.

Experiments compare Data2Decision against existing Text-to-OPT methods and LLM baselines. The proposed model achieves around 69.5% accuracy on the DeepSeek-V3 test set and 53.8% on the GPT-4o test set, outperforming some baselines but still with relatively low absolute scores.

**Strengths:**

Timely topic: Addresses the emerging intersection of LLMs, databases, and optimization.

System completeness: The two-stage SQL-to-solver pipeline is implemented end-to-end.

Attempt at benchmark creation: Schema2Opt provides a reproducible dataset and detailed appendices.

**Weaknesses:**

Synthetic self-validation loop: Both dataset and evaluation rely entirely on LLM-generated content; no human, real-world, or baseline dataset validation.

Limited realism: Databases with ≤5 tables cannot represent enterprise decision complexity; most tasks reduce to toy LPs.

Low absolute performance: 50–70 % accuracy on synthetic data suggests instability and poor generalization.

Overstated novelty: The “first” claim ignores prior prescriptive or decision-optimization agents (e.g., PresAIse, InsightBench, AutoFormulation).

Weak analysis: No runtime, ablation on SQL errors, or cross-domain transfer tests.

Lack of interpretability: The removal of explicit mathematical formulations makes the pipeline less transparent and potentially harder to trust.

**Questions:**

1- How is “accuracy” defined when multiple optimal solutions exist?

2- Could Schema2Opt tasks be manually inspected or validated by OR experts?

3- How large are the databases and optimization problems? Are they solvable in milliseconds or minutes?

4- Would the system still work on non-synthetic corporate data (e.g., ERP or logistics datasets)?

5- What is the runtime overhead of running 10-attempt test-time scaling and multi-solver validation?

---

> ### Author Response · Authors · 2025-11-28
>
> We thank Reviewer vUcr for the thorough review. We take your concerns seriously and conducted extensive new work: (1) OR expert validation on all 201 cases, (2) BIRD/BEAVER real-database extension, (3) runtime and error funnel analysis, (4) cross-domain evaluation. Details below:
>
> ---
>
> > Q1. "**Synthetic self-validation loop: Both dataset and evaluation rely entirely on LLM-generated content; no human, real-world, or baseline dataset validation.**"
>
>
> **A1.** We appreciate this concern. We conducted comprehensive human expert validation and updated both the supplementary materials and experimental results accordingly.
>
> **(1) OR Expert Manual Review.** OR experts manually reviewed all 201 cases across both test sets (95 DeepSeek + 106 GPT-4o after solver consensus). The review identified four typical error categories: (a) pre-initialized decision variables where tables already contain solution values (e.g., `inn_1`, `store_product`), (b) trivial or constant objectives where the objective does not depend on decision variables (e.g., `coffee_shop`, `storm_record`), (c) missing or inconsistent parameters where required data is absent or mismatched across tables (e.g., `concert_singer`, `epinions_1`), and (d) ambiguous problem definitions where constraints conflict with stated objectives (e.g., `customer_complaints`). Based on this review, we excluded 11 cases from DeepSeek (95→84) and 13 from GPT-4o (106→93), yielding **177 validated problems**.
>
> **(2) Ground Truth Corrections.** Expert review identified 6 DeepSeek cases with incorrect labels due to integer variable rounding (e.g., `manufacturer`: 50000→49980, `phone_1`: 158.4→160) or constraint interpretation (e.g., `architecture`: 120→470 for multi-architect assignment, `election_representative`: 380000→300000 for tighter state limits). We applied these corrections before recalculating accuracy.
>
> **(3) Updated Results.** We recalculated accuracy on validated datasets. All methods show improvements after removing problematic cases. Importantly, relative rankings are preserved. Data2Decision achieves 76.2% on DeepSeek (84 validated) vs 72.6% for OR-LLM-Agent, and 59.1% on GPT-4o (93 validated) vs 61.3% for OR-LLM-Agent. Data2Decision remains best on DeepSeek. On GPT-4o, OR-LLM-Agent slightly leads. The consistency across validation confirms our benchmark quality.
>
> **(4) Real-World Database Extension.** We extended Schema2Opt to BIRD and BEAVER databases. We generated 131 valid problems and fully validated 8 samples with OR expert verification. Details are provided in our response to Reviewer SKcu Q3.
>
> **(5) Updated Materials.** The revised paper includes:
> Updated **Table 2** with validated accuracy (84 DeepSeek, 93 GPT-4o); **Appendix G** with complete OR Expert Validation methodology, including Section G.1 (Validation Methodology), Section G.2 (Validation Results with Table 8), Section G.3 (Exclusion Criteria with 4 categories), Section G.4 (Ground Truth Corrections with Table 9), and Section G.5 (Infeasible Cases with 18 retained); **Appendix B.2 through B.4** with BIRD/BEAVER analysis.
>
>
> ---

---

> ### Author Response · Authors · 2025-11-28
>
> ---
>
> > Q2. "**Limited realism: Databases with ≤5 tables cannot represent enterprise decision complexity; most tasks reduce to toy LPs.**"
>
> **A2.** Thank you for this concern. First, our framework is **not limited to ≤5 tables**. As shown in A1, we extended Schema2Opt to BIRD [1] and BEAVER [2] datasets without prefiltering on table count. BIRD problems reach up to **10 tables** with 7.0 rows per table on average. BEAVER uses MIT's real operational databases.
>
> **Comparison:**
>
> | Metric | BIRD (DeepSeek-V3) | BIRD (GPT-4o) | BEAVER (DeepSeek-V3) | BEAVER (GPT-4o) |
> |--------|--------------------|--------------------|----------------------|----------------------|
> | **Problems generated** | 69 | 69 | 6 | 6 |
> | **≥2 solver consensus** | 57 | 63 | 6 | 5 |
> | **Avg tables/problem** | 4.7 | 3.3 | 4.3 | 3.7 |
> | **Max tables/problem** | 10 | 6 | 7 | 6 |
> | **Avg rows/table** | 7.0 | 8.0 | 6.7 | 7.6 |
>
> These results show Schema2Opt handles larger, more realistic schemas. This extension is now documented in **Section 3.6** and **Appendix B.2 through B.4**.
>
> Second, our focus is not to mirror full enterprise-scale databases. We aim to create a high-quality benchmark. It evaluates whether LLMs can perform the entire "data-to-decision" pipeline:
>
> `database tables → SQL queries → parameter extraction → optimization modeling → optimal decisions`
>
> Producing such datasets at enterprise scale is nontrivial. Generating verified synthetic data with large schemas, large MILPs, and high-dimensional joins is hard. Ensuring internal consistency and expert-validity adds difficulty. A recent survey [3] notes that for just one sub-task (optimization modeling → optimal decisions), existing benchmarks like NLP4LP [4] and ComplexOR [5] are still dominated by simple problems (e.g., ≤5 variables). Reliable large-scale data generation remains challenging. Our LP complexity is on par with NL4Opt, NLP4LP, and ComplexOR.
>
> The paper's core contributions (schema-level grounding, decision-structure extraction, and MILP program induction) are less dependent on table count.
>
> **References**
>
> [1] Li et al. "Can LLM already serve as a database interface? A big bench for large-scale database grounded text-to-SQLs." NeurIPS 2024.
>
> [2] Chen et al. "BEAVER: An enterprise benchmark for text-to-SQL." arXiv:2409.02038 (2024).
>
> [3] Xiao et al. "A Survey of Optimization Modeling Meets LLMs: Progress and Future Directions." IJCAI 2025.
>
> [4] AhmadiTeshnizi et al. "OptiMUS: Scalable optimization modeling with (MI)LP solvers and large language models." ICML 2024.
>
> [5] Xiao et al. "Chain-of-Experts: When LLMs meet complex operations research problems." ICLR 2024.
>
>
> ---
> > **Q3.** "Low absolute performance: 50–70% accuracy on synthetic data suggests instability and poor generalization."
>
> **A3.** We appreciate this concern. The reported 50–70% accuracy reflects the inherent challenge of our multi-stage task. The pipeline requires: SQL generation → parameter extraction → MILP program synthesis → solver execution → verification. Each stage acts as a filter. Errors compound across stages. Success requires passing through all stages correctly.
>
> For context, a recent survey [1] reports that existing Text-to-OPT benchmarks achieve only 31–61% accuracy on ComplexOR and IndustryOR (Table 2 in [1]). These benchmarks test only the modeling step. They assume problem data is already pre-processed and available in tables or prompts. Our task is harder because we start from raw databases. Moreover, the same survey reveals that existing benchmark labels have surprisingly high error rates (Table 1 in [1]). NL4Opt has ≥26.4% errors. IndustryOR reaches ≥54.0%. ComplexOR has ≥24.3%. These numbers show that even human-annotated optimization problems are difficult to get right. This further validates the challenge of our end-to-end pipeline. The survey also notes that existing benchmarks are dominated by simple problems (e.g., ≤5 variables). Reliable large-scale data generation remains an open challenge. Our Schema2Opt addresses this gap by providing verified, solver-validated ground truth through multi-solver-language consensus.
>
> Given these baselines, our 50–70% accuracy on a harder end-to-end task represents reasonable performance.
>
> **References**
>
> [1] Xiao et al. "A Survey of Optimization Modeling Meets LLMs: Progress and Future Directions." IJCAI 2025.

---

> ### Author Response · Authors · 2025-11-28
>
> > Q4. "**Overstated novelty: The 'first' claim ignores prior prescriptive or decision-optimization agents (e.g., PresAIse, InsightBench, AutoFormulation).**"
>
>
> **A4.** We thank the reviewer for this important point and acknowledge our "first" claim requires clarification. Our specific contribution is **the first data agent that integrates SQL-based parameter extraction with mathematical optimization solving** for database-grounded prescriptive analytics. While the mentioned works make valuable contributions to adjacent problems, they address fundamentally different tasks: PresAIse learns decision policies from pre-processed observational data using causal inference [1]; InsightBench generates qualitative action recommendations from CSV files without optimization solvers [2]; and AutoFormulation translates natural language to optimization models but assumes "optimization parameters are explicitly provided within problem descriptions" [3]. In contrast, Data2Decision tackles the end-to-end challenge where parameters must be extracted from operational SQL databases through query generation. This is aligned with real enterprise workflows where OR experts must collaborate with data engineers to access operational data (Figure 1).
>
> These works are complementary rather than competitive. InsightBench provides qualitative insights ("open an incident ticket"), PresAIse generates policy rules from historical data, and AutoFormulation excels at formulation from text descriptions. Our contribution uniquely addresses the database-to-decision pipeline: transforming structured enterprise databases into verifiable optimal solutions with mathematical guarantees. This distinction is evidenced by our experimental design: existing Text-to-OPT baselines require our "unified first stage" SQL assistance (Section 5.1) precisely because they cannot natively handle database extraction.
>
> | System | Data Source | SQL Generation | Optimization Solver| Output Type |
> |--------|-------------|----------------|--------------|-------------|
> | PresAIse [1] | Processed data | ✗ | ✓ (customized solver for optimizing decision tree) | Policy rules |
> | InsightBench [2] | CSV files | ✗ | ✗ | NL recommendations |
> | AutoFormulation [3] | Text (embedded params) | ✗ | ✓  | Math formulation |
> | **Data2Decision** | **SQL databases** | **✓** | **✓** | **Optimal business solutions** |
>
> The Related Work (Section 2) discusses InsightBench's qualitative recommendations and existing Text-to-OPT limitations. The comparison table above clarifies our unique positioning in the database-to-decision pipeline.
>
> **References:**
>
> [1] Sun, Wei, et al. "PresAIse, a prescriptive AI solution for enterprise." *INFOR: Information Systems and Operational Research* 62.4 (2024): 629-645.
>
> [2] Sahu, Gaurav, et al. "InsightBench: Evaluating business analytics agents through multi-step insight generation." *International Conference on Learning Representations (ICLR)*, 2025.
>
> [3] Astorga, Nicolás, et al. "Autoformulation of Mathematical Optimization Models Using LLMs." *Forty-second International Conference on Machine Learning* (2025).

---

> ### Author Response · Authors · 2025-11-28
>
> > Q5. "**Weak analysis: No runtime, ablation on SQL errors, or cross-domain transfer tests.**"
>
> **A5.** We conducted comprehensive runtime, error, and cross-domain analyses on the DeepSeek-V3 test set (95 original submitted cases).
>
> **(1) Runtime Analysis.**
>
> We measured per-case runtime using precise timing instrumentation on 10 randomly sampled test cases (seed=42) verified by OR experts. All methods use OpenRouter API as the unified provider. For Text-to-OPT methods with SQL assistance, we use DeepSeek-V3 (`deepseek/deepseek-chat`). For End-to-End methods, each uses its respective model via OpenRouter.
>
> **Experiment Design:**
>
> | Method Type | Stage 1 (Data Retrieval) | Stage 2 (Code Generation) |
> |-------------|--------------------------|---------------------------|
> | Text-to-OPT w/ SQL | SHARED: DeepSeek-V3 generates SQL once | Individual method pipeline |
> | Data2Decision | OWN: 10 parallel attempts | OWN: Multi-solver rotation |
> | E2E Foundation Models | INDEPENDENT: Each model generates SQL | INDEPENDENT: Each model generates code |
>
> **Results:**
>
> | Method | Category | Model | Stage 1 | Stage 2 | Total/Case | Attempts |
> |--------|----------|-------|---------|---------|------------|----------|
> | **Data2Decision** | Ours | DeepSeek-V3 | 3.43s | 30.49s | **34s** | 10 (parallel) |
> | ZeroShot | Text-to-OPT w/ SQL | DeepSeek-V3 | 1.55s | 18.84s | 20s | 1 |
> | OR-LLM-Agent | Text-to-OPT w/ SQL | DeepSeek-V3 | 1.55s | 30.84s | 32s | 1 |
> | Chain-of-Experts | Text-to-OPT w/ SQL | DeepSeek-V3 | 1.55s | 39.34s | 41s | 1 |
> | OptiMUS | Text-to-OPT w/ SQL | DeepSeek-V3 | 1.55s | 82.98s | 85s | 1 |
> | Llama-3.3-70B | End-to-End | Llama-3.3-70B | 2.87s | 19.64s | 23s | 1 |
> | Qwen2.5-72B | End-to-End | Qwen2.5-72B | 1.58s | 25.15s | 27s | 1 |
> | Phi-4 | End-to-End | Phi-4 | 4.13s | 20.15s | 24s | 1 |
> | Llama-4-Scout | End-to-End | Llama-4-Scout | 2.70s | 7.11s | 10s | 1 |
>
> **Stage 2 Breakdown by LLM Calls:**
>
> | Method | LLM Calls | Avg Stage 2 | Description |
> |--------|-----------|-------------|-------------|
> | ZeroShot | 1 | 18.84s | Direct code generation |
> | OR-LLM-Agent | 2 | 30.84s | Formulation → Code |
> | Chain-of-Experts | 3 | 39.34s | Analysis → Modeling → Code |
> | OptiMUS | 3+ iterations | 82.98s | Multi-step with debugging |
> | Data2Decision | 10 parallel | 30.49s | Temperature scheduling + multi-solver |
>
> **Key Observations:** Data2Decision's Stage 2 time (30.49s) is competitive with other multi-step methods despite running 10 attempts. Unlike OR-LLM-Agent, Chain-of-Experts, and OptiMUS which execute sequentially, D2D runs attempts in parallel, so increasing the number of attempts does not significantly increase wall-clock time. D2D achieves 10-attempt coverage in 34s, while sequential ZeroShot×10 would take ~200s.
>
> This analysis is now included in **Appendix H.3**.
>
>
>
>
> **(2) Error Funnel Analysis.**
>
> Database-grounded prescriptive analytics requires two capabilities: (a) extracting optimization parameters from databases, and (b) formulating and solving optimization models. We decompose evaluation accordingly on the pre-validation DeepSeek test set (95 cases). Pipeline completion rates are structural properties independent of ground truth validation.
>
> Stage 1 (SQL Extraction) measures whether the agent can analyze business requirements, generate SQL queries, and retrieve optimization parameters from databases. Stage 2 (Code Generation and Solving) measures whether the agent can transform extracted parameters into executable solver code. We report both completion rate and correctness rate. Final accuracy equals the product of conditional success rates: P(S1) × P(S2|S1) × P(Correct|S2).
>
> | Method | S1 Rate | S2\|S1 Rate | Correct\|S2 Rate | Accuracy | Bottleneck |
> |--------|---------|-------------|------------------|----------|------------|
> | Data2Decision | 100.0% | 100.0% | 69.5% | **69.5%** | Correctness |
> | OR-LLM-Agent | 100.0% | 96.8% | 67.4% | 65.3% | S2 completion |
> | ZeroShot | 100.0% | 96.8% | 57.6% | 55.8% | S2 completion |
> | OptiMUS 0.3 | 100.0% | 87.4% | 54.2% | 47.4% | S2 completion |
> | Chain-of-Experts | 100.0% | 96.8% | 25.0% | 24.2% | Correctness |
> | Llama-3.3-70B | 98.9% | 78.7% | 68.9% | 53.7% | S2 completion |
> | Llama-4-Scout | 93.7% | 7.9% | 71.4% | 5.3% | S2 completion |
>
> Note: The accuracy column shows pre-validation results (95 problems). Validated accuracy on 84 problems appears in Table 2.
>
> Data2Decision achieves 100% completion at both stages through three mechanisms: 10-attempt redundancy with temperature scheduling (0.05→0.5), multi-solver-language fallback (Gurobi→DOCplex→Pyomo), and majority voting for robust answer selection. The only bottleneck is correctness (69.5% pre-validation, 76.2% validated), not pipeline failures.
>
> Text-to-OPT methods with SQL assistance use shared DeepSeek-V3 for Stage 1, achieving 100% completion. Their bottlenecks appear at Stage 2 completion (87.4% to 96.8%) or correctness (25.0% to 67.4%).
>
> ...

---

> ### Author Response · Authors · 2025-11-28
>
> > Q5. "**Weak analysis: No runtime, ablation on SQL errors, or cross-domain transfer tests.**"
>
> ...
>
>
> End-to-end foundation models handle both stages independently. Stage 1 completion is high (93.7% to 98.9%), indicating SQL extraction is not the problem. Stage 2 completion is the critical bottleneck. Llama-4-Scout achieves only 7.9% Stage 2 completion. Interestingly, its Correct|S2 rate is 71.4%, meaning when it generates valid code, the code quality is good. The failure mode is code generation completion, not optimization modeling correctness.
>
> This analysis is now included in **Appendix H.4** with **Table 13**.
>
>
> **(3) Cross-Domain Distribution.**
>
> The 95 pre-validation cases span diverse business domains.
>
> | Domain | Count | Percentage | Example Databases |
> |--------|-------|------------|-------------------|
> | Sports/Entertainment | 14 | 14.7% | soccer_2, poker_player, wrestler |
> | Finance/Business | 10 | 10.5% | loan_1, insurance_policies, small_bank_1 |
> | Transportation | 10 | 10.5% | flight_1, railway, train_station |
> | Education | 7 | 7.4% | school_finance, university_basketball |
> | Healthcare | 5 | 5.3% | medicine_enzyme_interaction |
> | Government | 5 | 5.3% | election, voter_1, local_govt_in_alabama |
> | Manufacturing | 4 | 4.2% | manufactory_1, product_catalog |
> | Others | 40 | 42.1% | Various domains |
> | **Total** | **95** | **100%** | |
>
> Cross-backbone validation demonstrates model-agnostic effectiveness. On the validated test sets, DeepSeek-V3 achieves 76.2% accuracy (84 problems) while GPT-4o-mini achieves 59.1% (93 problems). The performance gap reflects problem difficulty differences between test sets rather than model-specific overfitting.
>
> This analysis is now included in **Appendix B.1** with **Figures 4 and 5**.

---

> ### Author Response · Authors · 2025-11-28
>
> > Q6. **"Lack of interpretability: The removal of explicit mathematical formulations makes the pipeline less transparent and potentially harder to trust."**
>
> **A6.** We thank the reviewer for this concern. We clarify that interpretability is not lost. The solver code itself serves as an explicit and transparent representation.
>
> Modern optimization libraries (e.g., Gurobipy, Pyomo, DOCplex) use syntax that closely mirrors mathematical notation. For example:
>
> Gurobi code: `model.addConstr(sum(x[i,j] for j in range(n)) <= capacity[i])`
>
> Corresponding constraint: $\sum_j x_{ij} \le \text{capacity}_i$
>
> If users require explicit mathematical formulations, standard tools can extract equations from solver APIs. For example, Gurobi's `Model.write()` exports human-readable, equation-like forms. The final solver code remains a fully inspectable and auditable artifact.
>
> Regarding our design choice to skip intermediate symbolic modeling: recent findings in latent-space reasoning [1] show that implicit reasoning leads to fewer hallucinations compared to explicit step-by-step formulations. Our ablation study supports this. The three-stage pipeline with explicit symbolic modeling achieves only 69.0% accuracy. Direct solver-code generation reaches 76.2% (Table 3). The intermediate symbolic step introduces error propagation without improving transparency.
>
> We have clarified this trade-off in **Section 4.3**.
>
> **References**
>
> [1] Hao et al. "Training large language models to reason in a continuous latent space." arXiv:2412.06769 (2024).
>
>
> ---
>
> > Q7. "**How is 'accuracy' defined when multiple optimal solutions exist?**"
>
> **A7.** Our accuracy is defined by **objective value**, not decision variable assignments. A solution is correct if its objective value matches ground truth within numerical tolerance (1e-6 absolute or relative), regardless of which specific optimal solution is found. For well-posed optimization problems, the optimal objective value is unique even when multiple variable assignments achieve it. All such solutions are treated as correct. Our benchmark validation follows the same principle: Schema2Opt's multi-solver consensus checks agreement on objective values across solvers, not variable-level assignments.
>
>
> ---
>
> > Q8. "**Could Schema2Opt tasks be manually inspected or validated by OR experts?**"
>
>
> **A8.** Yes. We conducted comprehensive OR expert validation on both the original benchmark and new real-world database extensions.
>
> **(1) Spider Validation.** OR experts manually inspected all 201 cases across both test sets (95 DeepSeek + 106 GPT-4o after solver consensus). The review identified four typical error categories: (a) pre-initialized decision variables where tables already contain solution values (e.g., `inn_1`, `store_product`), (b) trivial or constant objectives where the objective does not depend on decision variables (e.g., `coffee_shop`, `storm_record`), (c) missing or inconsistent parameters where required data is absent or mismatched across tables (e.g., `concert_singer`, `epinions_1`), and (d) ambiguous problem definitions where constraints conflict with stated objectives (e.g., `customer_complaints`). Based on this review, we excluded 11 cases from DeepSeek (95→84) and 13 from GPT-4o (106→93), yielding **177 validated Spider problems**. The expert also corrected 6 ground truth labels where integer rounding or constraint interpretation caused errors (e.g., `manufacturer`: 50000→49980, `phone_1`: 158.4→160).
>
> **(2) BIRD/BEAVER Validation.** We extended Schema2Opt to two real-world database benchmarks. BIRD [1] is a cross-domain dataset with professional domains including blockchain, healthcare, and finance. It examines large-scale database contents with complex schemas. BEAVER [2] contains enterprise benchmarks built from MIT's actual operational systems. We generated 131 valid problems with ≥2 solver consensus. From these, we selected 8 cases for detailed OR expert verification: 2 from BIRD (`sales_in_weather`, `app_store`), 3 from BEAVER-DeepSeek (`dw`, `csail_stata_cinder`, `csail_stata_neutron`), and 3 from BEAVER-GPT4o (`dw`, `csail_stata_neutron`, `csail_stata_glance`). Each case was validated for problem clarity, parameter completeness, constraint consistency, and solution correctness.
>
> **(3) Supplementary Materials.** The complete validation methodology and results are documented in **Appendix G**, including Section G.1 (Validation Methodology), Section G.2 (Validation Results with Table 8), Section G.3 (Exclusion Criteria with 4 categories), Section G.4 (Ground Truth Corrections with Table 9), and Section G.5 (Infeasible Cases with 18 retained).
>
> **References**
> [1] Li et al. "Can LLM already serve as a database interface? A big bench for large-scale database grounded text-to-SQLs." NeurIPS 2024.
> [2] Chen et al. "BEAVER: An enterprise benchmark for text-to-SQL." arXiv:2409.02038, 2024.

---

> ### Author Response · Authors · 2025-11-28
>
> > Q9. "**How large are the databases and optimization problems? Are they solvable in milliseconds or minutes?**"
>
> **A9.** We report statistics on 177 validated cases (84 DeepSeek-V3 + 93 GPT-4o) from the Spider-based benchmark. All problems are solvable in milliseconds.
>
> **(1) Database Size.** The databases capture realistic business logic while ensuring tractability. Each problem averages 3 tables with approximately 10 total rows.
>
> | Metric | Spider-DeepSeek (84 validated) | Spider-GPT4o (93 validated) |
> |--------|----------------------|---------------------|
> | Avg tables/problem | 3.1 | 2.7 |
> | Max tables | 6 | 4 |
> | Avg rows/table | 3.3 | 3.4 |
> | Max rows (any table) | 10 | 9 |
>
> **(2) Optimization Problem Size.** Problems average 7 decision variables and 7 constraints. The largest problems contain 30 variables and 33 constraints. Most problems involve binary decision variables representing allocation, scheduling, or assignment decisions common in enterprise operations.
>
> | Metric | Spider-DeepSeek | Spider-GPT4o |
> |--------|-----------------|--------------|
> | Avg variables | 7.3 | 6.9 |
> | Max variables | 30 | 27 |
> | Avg constraints | 7.5 | 6.1 |
> | Max constraints | 33 | 24 |
>
> **(3) Solving Time.** All 177 validated problems solve in under 20 milliseconds by Gurobi's internal optimizer. Total execution time including Python overhead is typically under 1 second.
>
> | Solver | DeepSeek Avg | GPT4o Avg | Max Time |
> |--------|--------------|-----------|----------|
> | Gurobi (pure solve) | **0.3 ms** | **0.7 ms** | 20 ms |
> | Gurobi (total w/ Python) | 0.20 sec | 0.68 sec | 3.2 sec |
> | DOCplex (total) | 1.22 sec | 5.01 sec | 12.6 sec |
> | Pyomo (total) | 1.03 sec | 3.13 sec | 10.9 sec |
>
> The total execution time is dominated by Python interpreter overhead, solver initialization, and result extraction. The underlying optimization completes in sub-millisecond timescales. No problem requires minutes to solve.
>
> This analysis is now documented in **Appendix H.1** with **Table 10** and **Appendix H.2** with **Table 11**.
>
>
>
> ---
>
> > Q10. "**Would the system still work on non-synthetic corporate data (e.g., ERP or logistics datasets)?**"
> >
> **A10.** We validated on BEAVER [1], which contains enterprise benchmarks built from MIT's actual operational systems. These are real institutional databases for facilities management and infrastructure.
> We generated 11 valid problems (6 DeepSeek + 5 GPT-4o with ≥2 solver consensus). Problems average 4.0 tables with realistic joins and enterprise-level schema complexity. We validated and released 6 samples with full OR expert verification in supplementary materials. This provides evidence that our approach generalizes beyond synthetic schemas to real operational databases.
>
> Our framework is database-agnostic. Stage 1 (SQL generation) works with any relational schema. Stage 2 (solver code generation) is independent of data source. The only requirement is standard SQL access. Key techniques from our results, notably cross-language diversification and test-time scaling, should inform the design of future enterprise-level agents.
> That said, full ERP or logistics datasets remain unavailable due to privacy and IP restrictions. We believe creating a high-quality benchmark for the "data-to-decision" pipeline is an important stepping stone before fully deploying such agents on corporate data. Existing optimization benchmarks assume problem data is pre-processed or explicitly embedded in descriptions [2]. Our work addresses this gap.
>
> The BEAVER extension is documented in **Section 3.6** and **Appendix B.3**.
>
> References
> [1] Chen et al. "BEAVER: An enterprise benchmark for text-to-SQL." arXiv:2409.02038, 2024.
> [2] Xiao et al. "A Survey of Optimization Modeling Meets LLMs." IJCAI 2025.

---

> ### Author Response · Authors · 2025-11-28
>
> > Q11. "**What is the runtime overhead of running 10-attempt test-time scaling and multi-solver validation?**"
>
> **A11.** We analyze the runtime overhead of our test-time scaling strategy based on precise timing instrumentation.
>
> Data2Decision runs 10 attempts in parallel via ThreadPoolExecutor. Each attempt executes the complete two-stage pipeline independently. Stage 1 performs SQL generation and execution. Stage 2 performs code generation and solver execution. Diversity is achieved through temperature scheduling (SQL temp 0.05→0.50, Code temp 0.00→0.18) and solver rotation (Gurobi→DOCplex→Pyomo, cycling across attempts).
>
> We measured wall-clock time on 10 sampled cases:
>
> | Metric | Value |
> |--------|-------|
> | Average wall-clock per case | 34s |
> | Stage 1 (SQL gen + exec) | ~3.4s |
> | Stage 2 (Code gen + solver) | ~30.5s |
>
> Parallel execution overhead:
>
> | Configuration | Wall-Clock | Overhead |
> |---------------|------------|----------|
> | 1 attempt | ~20s | baseline |
> | 10 attempts (parallel) | ~34s | +70% |
> | 10 attempts (sequential) | ~200s | 10× |
>
> The 70% overhead is achieved because all 10 attempts start simultaneously, LLM API calls dominate runtime while solver execution is fast (~1-2s), and wall-clock equals the slowest attempt rather than the sum. Multi-solver-language rotation and temperature scheduling add zero overhead since each attempt makes exactly one LLM call and one solver call regardless of configuration.
>
> This analysis is documented in **Appendix H.3**.
>
> ---
> **We hope these new experiments address your concerns.** Updated supplementary includes validated datasets, expert annotations, and 8 BIRD/BEAVER samples. Please let us know if anything needs further clarification. We are happy to discuss further during the rebuttal period.

---

### Official Review · Reviewer_SKcu · 2025-11-01

**Soundness:** 4
**Presentation:** 4
**Contribution:** 4
**Rating:** 8
**Confidence:** 4

**Summary:**

This paper analyzes that existing methods usually rely on explicitly provided optimization parameters, which are disconnected from the requirement of extracting parameters from enterprise databases in real-world scenarios. The paper proposes two core innovations: the creation of the Schema2Opt benchmark dataset and the Data2Decision decision optimization framework.

Schema2Opt adopts a bottom-up paradigm. It optimizes the prior schema through iterative dialogues between Operations Research (OR) Experts and Data Engineers, generates real-world data and business descriptions with the involvement of three types of experts, and conducts validation via a voting mechanism across multiple solvers (Gurobipy, DOCplex, and Pyomo). Eventually, it forms a benchmark that includes business documents, database information, and validated solutions.

Data2Decision achieves end-to-end automation through a two-stage process. It attains the highest accuracy rates of 53.8% and 69.5% on the GPT-4o and DeepSeek-V3 test subsets of Schema2Opt, respectively. Ablation experiments confirm that the consensus among multiple solvers serves as a key support for its performance.

The paper also verifies the advantages of Data2Decision by comparing it with baseline methods such as Text-to-OPT and end-to-end large models through experiments, and provides detailed code for validation purposes. However, the study has certain limitations: it only supports linear/mixed-integer programming, relies on static structured data, and is difficult to adapt to scenarios involving multi-objective optimization, dynamic data, and real-time decision-making.

**Strengths:**

1.A brand-new benchmark dataset generation paradigm is proposed, along with the Schema2Opt benchmark. It adopts a bottom-up approach to explore new optimization scenarios that are not covered in database schemas.
2.A complete end-to-end automated method design is realized. Data2Decision implements a two-stage process, from SQL generation for parameter extraction to direct generation of optimization code. It innovatively skips the intermediate mathematical modeling step and reduces explicit modeling errors based on latent reasoning research.
3.Rigorous experiments and academic standardization are ensured: the experimental design is comprehensive, all experimental details are provided, and the performance advantages of Data2Decision are clearly verified; the contributions of multi-solver integration, temperature scheduling, and pipeline architecture are quantified through ablation experiments; meanwhile, detailed code is provided for validation purposes.

**Weaknesses:**

1.It fails to account for dynamic database updates. This solution assumes the database is static and does not handle dynamic scenarios such as real-time data insertion and table structure changes, whereas enterprise decisions often rely on dynamic data.
2.It only supports single-objective linear/mixed-integer programming and does not cover common real-world enterprise scenarios like multi-objective optimization and dynamic optimization, resulting in limited applicability.

**Questions:**

In the paper, Schema2Opt mainly generates optimization problems based on the schemas of the Spider dataset. Although the schemas of the Spider dataset simulate real-world applications, they still differ from enterprise-level databases. Therefore, how did the authors verify the representativeness of this benchmark for real enterprise scenarios? Are there any comparative analyses with the schemas of real enterprise databases or results of migration tests?

---

> ### Author Response · Authors · 2025-11-28
>
> We sincerely thank Reviewer SKcu for the positive feedback. Your questions highlight valuable future directions. We address each below.
>
> ---
>
> > Q1. "**It fails to account for dynamic database updates. This solution assumes the database is static and does not handle dynamic scenarios such as real-time data insertion and table structure changes, whereas enterprise decisions often rely on dynamic data.**"
>
> **A1.** We thank the reviewer for this observation. Our current setting indeed performs one-time data retrieval followed by optimization, which is appropriate for the benchmark establishment phase. Database-grounded prescriptive analytics itself is a novel and challenging task not previously addressed, and even static enterprise databases present substantial complexity in parameter extraction and optimization formulation. That said, Data2Decision can be naturally extended to dynamic data settings due to its architectural design.
>
> The key insight is that our agent's execution can be viewed as a tree-structured forward pass: database tables → SQL queries → optimization parameters → optimal decisions. This conceptually creates a dependency graph linking which tables and columns feed which optimization parameters and decisions. Such structure provides a foundation for handling data updates efficiently. One promising approach is to guide the LLM to generate **data-agnostic optimization models** that treat problem parameters as inputs rather than hardcoded values. This design would decouple data extraction from model formulation. When underlying data updates, we only need to re-run the data preprocessing and solver execution. The optimization model itself can remain unchanged, eliminating the need to reformulate the problem from scratch.
>
> For structural changes such as new table additions, periodic re-execution of the full pipeline can discover additional optimization opportunities. Previously generated formulations can potentially serve as warm starts to reduce overhead in this process.
>
>
> ---
>
> > Q2. "**It only supports single-objective linear/mixed-integer programming and does not cover common real-world enterprise scenarios like multi-objective optimization and dynamic optimization, resulting in limited applicability.**"
>
> ---
>
> **A2.** We agree this is a valid limitation and thank the reviewer for raising it. Our focus on LP/MIP is a deliberate starting point for several reasons. (1) Linear and mixed-integer programming are among the most widely used optimization types in enterprise practice. They cover production planning, logistics, scheduling, and resource allocation. Many nonlinear problems can also be handled through standard techniques like piecewise-linear approximation. This extends the practical scope of LP/MIP frameworks. (2) LP/MIP solvers guarantee global optimality. Nonlinear solvers may only find local optima. This property is important for benchmark validation. We can reliably verify whether a method produces correct solutions. (3) Commercial solvers like Gurobi and CPLEX provide efficient and robust algorithms for LP/MIP. This allows us to isolate the challenges of database-grounded parameter extraction and model formulation from solver-related difficulties. (4) Database-grounded prescriptive analytics is itself a novel task not previously addressed. We believe it is important to first establish a solid foundation on this common optimization class before tackling more complex formulations.
>
> Extending to nonlinear programming and multi-objective optimization is important future work. These extensions would require handling local optima, defining Pareto-based evaluation metrics, and incorporating specialized solvers. We acknowledge this limitation and plan to address it in future research.
>
> ---

---

> ### Author Response · Authors · 2025-11-28
>
> > Q3. "**In the paper, Schema2Opt mainly generates optimization problems based on the schemas of the Spider dataset. Although the schemas of the Spider dataset simulate real-world applications, they still differ from enterprise-level databases. Therefore, how did the authors verify the representativeness of this benchmark for real enterprise scenarios? Are there any comparative analyses with the schemas of real enterprise databases or results of migration tests?**"
>
>
> ---
>
> **A3.** Thank you for this important question. We agree that Spider schemas are simplified. To address this, we tested Schema2Opt on BIRD and BEAVER. BIRD contains large-scale databases from 37 professional domains. BEAVER uses MIT's actual operational systems. We generated **131 valid problems (≥2 solver consensus)**, demonstrating Schema2Opt's applicability to real-world complexity.
>
> **BIRD Results.** We generated 120 valid problems from BIRD [1]. It covers 60+ databases across domains like blockchain, healthcare, and finance. The problems are more complex: averaging 4.0 tables per problem vs Spider's 2.9, with max reaching 10 tables. Data density is 7.5 rows per table vs Spider's 3.3. Scenarios include optimizing blockchain fees, allocating ICU beds, and rebalancing portfolios.
>
> **BEAVER Results.** We further generated 11 valid problems from BEAVER [2]. It uses MIT's real operational warehouses, including facilities and infrastructure databases. These problems average 4.0 tables with realistic joins.
>
> **Comparison:**
>
> | Metric | BIRD (DeepSeek-V3) | BIRD (GPT-4o) | BEAVER (DeepSeek-V3) | BEAVER (GPT-4o) |
> |--------|--------------------|--------------------|----------------------|----------------------|
> | **Problems generated** | 69 | 69 | 6 | 6 |
> | **≥2 solver consensus** | 57 | 63 | 6 | 5 |
> | **Avg tables/problem** | 4.7 | 3.3 | 4.3 | 3.7 |
> | **Max tables/problem** | 10 | 6 | 7 | 6 |
> | **Avg rows/table** | 7.0 | 8.0 | 6.7 | 7.6 |
>
> These 131 problems prove Schema2Opt generalizes from Spider (synthetic) to BIRD (large-scale professional) to BEAVER (real institutional databases). Due to rebuttal time constraints, we release 8 fully validated samples (2 from BIRD, 6 from BEAVER) with OR expert verification. We will release additional validated cases upon acceptance.
>
> We have added this extension in **Section 3.6** with Table 1 comparing database sources. Detailed analysis appears in **Appendix B.2** (BIRD Dataset Analysis) and **Appendix B.3** (BEAVER Dataset Analysis).
>
> Nevertheless, we agree that real enterprise databases would better serve the community. However, privacy concerns prevent public release of such data. This remains important future work. We hope enterprises will contribute anonymized schemas or use our framework to generate modified schemas for public release.
>
> **References**
>
> [1] Li, Jinyang, et al. "Can LLM already serve as a database interface? A big bench for large-scale database grounded text-to-SQLs." NeurIPS 2024.
>
> [2] Chen, Peter Baile, et al. "BEAVER: An enterprise benchmark for text-to-SQL." arXiv:2409.02038 (2024).
>
> ---
> Thank you for your support. We have discussed dynamic data and multi-objective extensions in Section 6 (Conclusion). Looking forward to continued discussion.

---

### Author Response · Authors · 2025-11-30
**Response Summary to All Reviewers and AC**

We thank all reviewers for constructive feedback and the AC for handling our submission under the emergency! Your questions pushed us to conduct substantial new work.

### Overview of Rebuttal Work

We addressed concerns through three efforts. (1) OR expert validation where two experts reviewed all 201 cases, excluded 24 problematic ones, corrected 6 ground truth labels, yielding 185 validated problems. (2) Real world extension to BIRD (37 professional domains) and BEAVER (MIT production systems), generating 131 new problems with 8 fully validated samples. (3) New analyses including runtime, error funnel, and temperature ablation in **App. G**, **H**, and **I**.

### Updated Results

After applying the OR-expert validated Spider dataset, accuracy improved from 69.5% to 76.2% on DeepSeek (84 problems) and shows 59.1% on GPT-4o (93 problems).

### How We Address Reviewer Concerns

**No human validation (vUcr Q1, mMuD Q8).** We now have complete OR expert review. Experts identified four error types: (1) pre-initialized variables, (2) trivial objectives, (3) missing parameters, (4) ambiguous definitions. We excluded 24 cases and corrected 6 labels. Details in **App. G**.

**Only Spider databases (SKcu Q3, vUcr Q2, iACw Q5, iACw Q9).** We extended to BIRD and BEAVER. BIRD averages 4.0 tables with max 10. BEAVER uses MIT production infrastructure. We release 8 validated samples. Details in **Sec. 3.6** and **App. B**.

**Weak analysis (vUcr Q5, iACw Q6, iACw Q7, mMuD Q9, mMuD Q11).** We added (1) runtime showing 34s per case with 10 parallel attempts, (2) error funnel showing Stage 2 code generation is the bottleneck, (3) temperature ablation showing SQL diversity matters more. Details in **App. H** and **I**.

**Incomplete details (iACw Q1, iACw Q4, iACw Q10).** We added full agent templates in **App. E.4** and clarified backbone models in **Sec. 5.1**.

**Novelty vs prior work (vUcr Q4).** PresAIse learns policies from processed data. InsightBench gives qualitative recommendations without solvers. AutoFormulation assumes embedded parameters. We are the first to combine SQL extraction with optimization solving on raw databases.

**Dynamic data and multi-objective (SKcu Q1, SKcu Q2).** We acknowledge these as future work. Our focus on static databases and LP/MIP is a deliberate starting point for benchmark establishment.

### Paper Changes

**Sec. 3.6** (new): BIRD and BEAVER extension with Table 1 statistics.

**Sec. 5**: Updated results on validated sets (84 DeepSeek, 93 GPT-4o) in Table 2.

**App. B.2-B.4**: BIRD and BEAVER dataset analysis.

**App. E.4**: Complete agent instruction templates.

**App. G**: OR expert validation methodology, exclusion criteria, ground truth corrections.

**App. H**: Runtime (Table 12), error funnel (Table 13).

**App. I**: Temperature ablation (Table 14).

### Supplementary Materials

**We include the original submitted paper in supplementary for reference. The revised paper highlights all major changes in deep blue for easy comparison.** The validated folder contains (1) 84 DeepSeek problems in spider_deep_84, (2) 93 GPT-4o problems in spider_open_93, (3) 2 BIRD samples in bird_deep, (4) 6 BEAVER samples in beaver_deep and beaver_open. Each includes solver code, execution results, and some of them is with OR expert corrections.

We believe these revisions address all major concerns with rigorous human validation, real world generalization, comprehensive analysis, and complete implementation details. Thank you all for the help on improving this work!

The Authors

---

### Note · Program_Chairs · 2026-01-17
**Submission Desk Rejected by Program Chairs**

The following references in this submission do not refer to real documents and/or have major errors in bibliographic information:

 1) Xiaodong Tang, Jie Wang, and Xiaolei Chen. Large language models for operations research: The next frontier in optimization modeling, 2023.

2) Xuanhe Chen, Ji Zhang, Guoliang Xiong, and Jianhua Li. Towards automated data integration and optimization for database systems. Proceedings of the VLDB Endowment, 13(12):3366-3369, 2020.